# TART: A plug-and-play Transformer module for task-agnostic reasoning

**Kush Bhatia**[*]        **Avanika Narayan**[*]        **Christopher De Sa**        **Christopher Ré**

{kushb, avanika, chrismre}@cs.stanford.edu, cdesa@cs.cornell.edu

## Abstract

Large language models (LLMs) exhibit in-context learning abilities which enable the same model to perform several tasks without any task-specific training. In contrast, traditional adaptation approaches, such as fine-tuning, modify the underlying models for *each* specific task. In-context learning, however, consistently underperforms task-specific tuning approaches *even* when presented with the same examples. While most existing approaches (e.g., prompt engineering) focus on the LLM's learned representations to patch this performance gap, our experiments actually reveal that LLM representations contain sufficient information to make good predictions. As such, we focus on the LLM's reasoning abilities and demonstrate that this performance gap exists due to their inability to perform simple probabilistic reasoning tasks. This raises an intriguing question: Are LLMs actually capable of learning how to reason in a task-agnostic manner? We answer this in the affirmative and, as a proof of concept, propose TART which generically improves an LLM's reasoning abilities using a synthetically trained reasoning module. TART trains this Transformer-based reasoning module in a task-agnostic manner using *only synthetic* logistic regression tasks and composes it with an arbitrary real-world pre-trained model without any additional training. With a single inference module, TART improves performance across different model families (GPT-NEO, PYTHIA, BLOOM), model sizes (100M - 6B), tasks (14 NLP classification tasks), and even across different modalities (audio and vision). On the RAFT Benchmark, TART improves GPT-NEO (125M)'s performance such that it outperforms BLOOM (176B), and is within $4\%$ of GPT-3 (175B). [2]

## 1   Introduction

Large language models (LLMs) show in-context learning capabilities which enable them to perform a task given only a few examples, without updating the model parameters [9, 7]. This task-agnostic capability allows for a single model to be applied to a wide range of tasks [1, 41, 28]. In contrast, traditional task adaptation approaches, such as fine-tuning, update the model parameters for each specific task.

Despite being task-agnostic, in-context learning is seldom the practitioner's method of choice since it consistently underperforms task-specific adaptation approaches [20, 9]. Most existing works attribute this performance gap to the limited context window of LLMs which can only accommodate a few task examples [15, 14, 23]. However, we show that this gap between in-context learning and fine-tuning approaches exists *even* when presented with the same task examples; in-context learning can underperform fine-tuning by up to 30 points across binary classification benchmarks (see Figure 5a).

We first investigate why this quality gap exists. We decompose an LLM's in-context learning capability into two abilities: learning good *representations* for the task and performing probabilistic inference,

---

[*]Equal Contribution

[2]Our code and model is available at `https://github.com/HazyResearch/TART`

37th Conference on Neural Information Processing Systems (NeurIPS 2023).

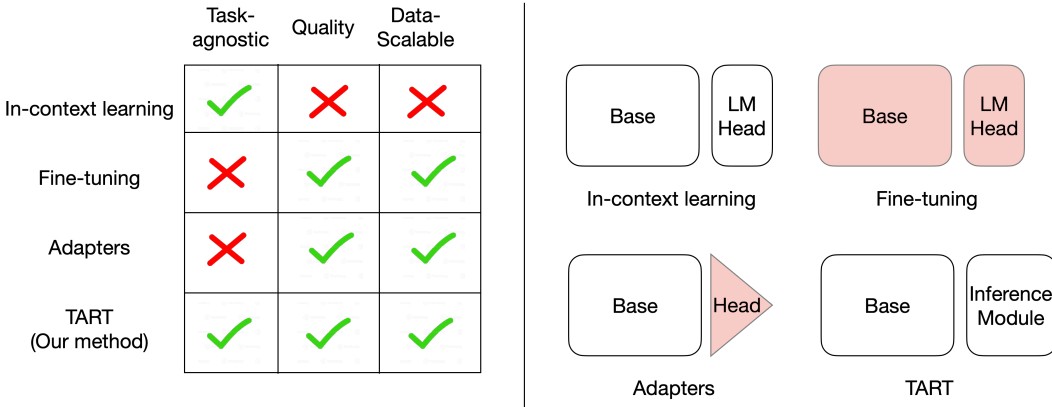

Figure 1: **Taxonomy of task adaptation strategies.** (Left) Comparison of different adaptation strategies across three desiderata: task-agnostic, quality, scalability. (Right) Demonstration of parameter updates across adaptations strategies, colored regions represent parameter changes as a result of the adaptation strategy.

or *reasoning*, over these representations[3]. Is the gap because the data representations learned by the models lack task-specific information or is there some deficiency in the the model's ability to use this information. We take a first step towards understanding this hypotheses experimentally in Section 2 by measuring both the reasoning and the representation gaps across a variety of LLM families (GPT-NEO [6], PYTHIA [5], BLOOM [33]) over a suite of binary classification tasks. Our results suggest that LLMs indeed have the necessary information in the representations, and that the majority of the quality gap (up to 79%) can be attributed to their inability to perform simple forms of reasoning, e.g. linear classification. We further find that adaptation methods (such as fine-tuning) improve the model along both representation and reasoning, but primarily improve the task-specific classification (or reasoning) ability which accounts for 72% of the gained performance on the task.

Rather surprisingly, most existing techniques for improving the performance gap, such as prompt engineering or active example selection, focus entirely on the LLM's learned representations. With these methods, the users are effectively searching the language space for a natural language template that best represents the task. This search enables the users to optimize the representations for their specific task which makes the LLM more likely to solve it. In contrast, our work explores the orthogonal direction of improving the LLM's reasoning abilities.

As a first step towards improving this reasoning ability, we fine-tune LLMs using synthetically generated binary classification tasks (see Section A.1 for examples). While this approach provides an improvement over the model's base in-context learning performance (up to 17%, see Figure 7), this approach requires fine-tuning each LLM individually and might affect their generative abilities [18]. This brings about a question: can we improve the reasoning capabilities of LLMs without interfering with it pre-existing capabilities, that is, can we make the process both task and model agnostic?

We show that it is indeed possible to improve the reasoning capabilities in a completely agnostic manner. Rather than directly fine-tuning, our proposed algorithm TART, Task-Agnostic Reasoning Transformer (Figure 2), improves an LLM's reasoning abilities using an independent synthetically trained module. TART trains this additional Transformer module using only synthetically generated binary classification tasks from the logistic model independent of the downstream task or the base LLM. This inference module can be composed, *without any additional training*, with the embeddings of an arbitrary pre-trained LLM to improve upon its reasoning abilities.

TART is *task, model, and domain* agnostic. Using a single inference module trained on synthetic data, we exhibit that TART not only generalizes across three model families (GPT-NEO, PYTHIA, BLOOM) over 14 NLP classification tasks, but even across different domains (vision and speech; see Figure 6). In terms of quality, we show that TART's performance is 18.4% better than in-context learning, 3.4% better than task-specific adapters, and is within 3.1% of full task-specific fine-tuning across a suite of NLP tasks. On the RAFT Benchmark [2], TART improves GPT-NEO (125M)'s performance such that it outperforms BLOOM (176B), and is within $4\%$ of GPT-3 (175B). TART is

---

[3]In Section 2.3, we precisely define the this representation-reasoning decomposition in eq. (1)

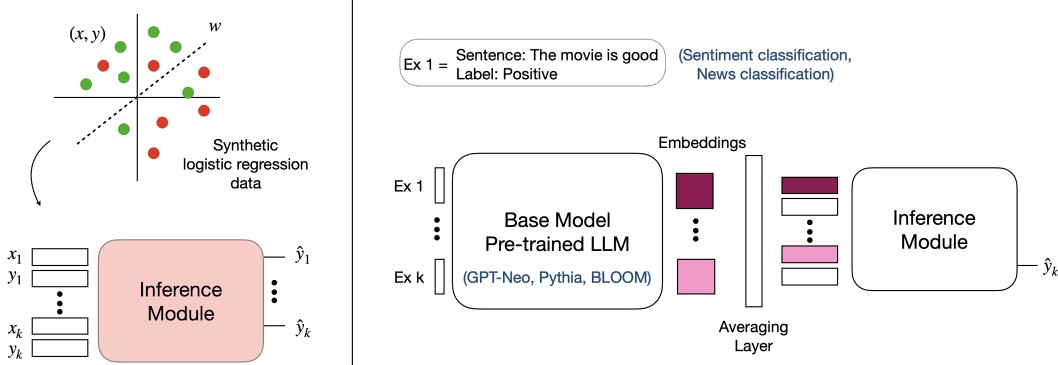

Figure 2: **TART**. (Left) Inference module training procedure: The inference module is trained on sequences of synthetically generated logistic regression tasks. (Right) End-to-end framework: TART composes a pre-trained LLM with the inference module. TART uses the LLM to embed the input text. These embeddings, along with the train labels, are passed as a sequence to the inference module which generates a final prediction.

data-scalable and overcomes the limited context length bottleneck of in-context learning. While each example spans multiple tokens in an LLM, often spanning hundreds of tokens, TART's reasoning module encodes each example using only two tokens – one for the context and the other for the label. This data-scalability can lead to improvements of up to $6.8\%$ (see Figure 5c).

From a theoretical standpoint, we show that the generalization abilities of TART depends mainly on the distribution shift between the natural text embedding distribution produced by the LLM and the synthetic data distribution, measured in terms of the Wasserstein-1 metric (Theorem 1 in Section 3.4).

To summarize, our main contributions are as follows:
- Study why in-context learning does not perform as well as task-specific fine-tuning despite having access to the same information, via a representation-reasoning decomposition.
- Propose a new task-agnostic method, TART, which bridges the performance gap to task-specific methods and is trained using only synthetic classification data.
- Demonstrate that TART works across different NLP tasks for a range of model families. The same inference module generalizes to vision and speech domains, suggesting that this capability is somehow uniformly lacking across a range of foundation models.

**Related work.** Prompt engineering focuses on improving the in-context task adaptation abilities of LLMs by modifying prompts. A line of work improves performance by carefully designing the natural language task specifications [4, 42] while others improve performance by optimizing the examples chosen for the prompt [10, 24], encouraging the models to sequentially reason [16, 42, 45] and aggregating prompts [39, 38]. Unfortunately, prompt-based task adaptation is noisy [26]. Alternatively, prompt tuning improves the in-context abilities of models by training a small amounts of learnable vectors [21, 20, 25] for specific tasks. While these methods have been shown to improve in-context learning performance, they require task-specific fine-tuning and are not task-agnostic.

Recent works seek to understand the in-context learning property of LLMs by presenting mechanistic interpretations of in-context learning [36] and performing exploratory analysis of in-context learning behaviors [43]. Existing literature demonstrates that LLMs can learn simple function classes in-context [11] and propose that LLMs are performing gradient descent when learning tasks in-context [36]. Complementary to these, our work provides insights on the mechanisms of in-context learning and its deficiencies. Furthermore, task transfer strategies adapt LLMs to a pre-specified target task. Strategies range from parameter efficient finetuning (PEFT) [12, 47] to Low-Rank adaptation (LoRA) [13] which introduces trainable rank decomposition matrices into each layer.

## 2 Task adaptation strategies: Taxonomy and evaluation

We begin by describing the problem of adapting pre-trained language models for a collection of downstream tasks while being task-agnostic, competent in performance, and data-scalable. Given

these criteria, we evaluate existing task adaptation approaches and propose a representation-reasoning decomposition to understand their relative performances.

## 2.1 Problem statement and evaluation criteria

Our focus is on methods for adapting pre-trained large language models (LLMs) for downstream tasks. Specifically, given an LLM and limited labeled data for a task, how does one adapt the model to the task? In this work, our focus is on classification tasks. For ease of presentation, we consider binary classification tasks in the main paper and defer the extension to multi-class setup to Appendix D. When evaluating a task adaptation strategy, we care about the following properties:

**Task-agnostic.** Given the general capabilities of pre-trained LLMs, we strive to utilize the same model across different tasks without requiring any task-specific training. With the increase in model sizes, the cost of deploying task-specific models increase both during training (expensive hyper-parameter search) as well as during inference (deploying several models). In general, task-agnostic methods will scale better with increasing model sizes by side-stepping both these costs.

**Performance quality.** We would like the adaptation approaches to be competitive in performance when compared with task-specific approaches across a wide range of tasks. For the binary classification tasks, the method should have accuracy comparable with task-specific approaches.

**Data-scalable.** The task adaptation method should be scalable with the number of labeled task examples. In particular, the method should be capable of learning from large datasets, and continually improve its performance quality.

## 2.2 Taxonomy of task adaptation strategies

We can broadly taxonomize the existing task adaptation strategies for LLMs as in-context learning, fine-tuning the model, and training task-specific adapters (see Figure 1).

**In-context learning.** In-context learning allows for adapting the model without updating any model parameters, by simply providing a few demonstrations of the task in the LLM prompt. In-context learning is completely task-agnostic since the same model can be used across tasks since no weights are updated at inference time. However, its performance is usually not on par with task-specific methods and it does not scale well with data since the number of examples that can be utilized is bottlenecked by the context length of the model.

**Fine-tuning.** This traditional class of methods update the model weights to adapt it specifically for the task, typically by performing gradient descent over the labeled dataset. Fine-tuning methods are not task-agnostic since they change the underlying model significantly but usually achieve state-of-the-art performance for any given task and are data scalable.

**Adapters.** Adapters adapt the underlying LLM to a specific task by composing the LLM base model with an additional set of parameters which are optimized for the task. In contrast to fine-tuning which performs updates to the base model, adapters keep the base model frozen and only update the additional parameters. Performance of adapters is usually competitive with full fine-tuning.

## 2.3 Understanding performance via Representation-Reasoning decomposition

From the taxonomy of task adaptation approaches, only in-context learning satisfies the task-agnostic property but it consistently underperforms the task-specific tuning approaches. This section investigates why this performance gap exists. We hypothesize that it is either because (a) the representations learned by the LLM are insufficient to learn a good predictor for the specific task, or (b) the LLM lacks the capability to reason over these representations to make good predictions for the task.

To understand whether the representations have sufficient information, we train a task-specific linear classifier using these representations, also known as linear probing, and evaluate its accuracy ($\text{Acc}_{\text{LR}}$). Using this as an intermediate, we decompose the performance gap

$$\Delta_{\text{perf}} := \text{Acc}_{\text{FT}} - \text{Acc}_{\text{ICL}} = \underbrace{\text{Acc}_{\text{FT}} - \text{Acc}_{\text{LR}}}_{\Delta_{\text{rep}}} + \underbrace{\text{Acc}_{\text{LR}} - \text{Acc}_{\text{ICL}}}_{\Delta_{\text{reas}}} \tag{1}$$

where $\Delta_{\text{rep}}$ represents the gap in performance which can be attributed to insufficient representation capacity and $\Delta_{\text{reas}}$ is the performance gap due to insufficient reasoning abilities. Mathematically, we define the quality of a models representations to be the gap between the fine-tuned model's accuracy

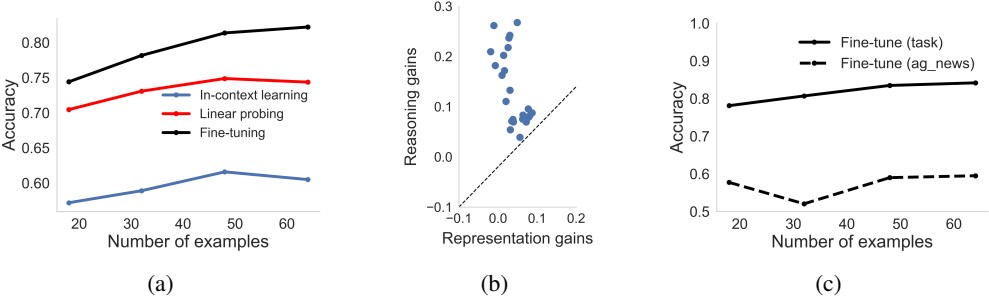

Figure 3: All results for GPT-Neo (125M). (a) Accuracy of in-context learning vs. linear probing on model embeddings: representations have sufficient information. (b) Fine-tuning majorly improves task-specific reasoning across datasets. (c) Accuracy (averaged across 6 datasets) of task-specific fine-tuned model vs. accuracy of model fine-tuned on AGNews and evaluated on task. Fine-tuning hurts task-agnosticity.

and the accuracy of linear probing – the smaller the gap the better the representations. Similarly, the models reasoning ability, or the ability to perform linear classification, is measured by the gap between the in-context learning and linear probing accuracies – if this gap is small, that means the model does not improve by learning any task-specific decision boundary and therefore already possesses the ability to reason.

Using the above decomposition, we consider the following hypotheses:

H1. LLM representations have enough information to perform the task in-context, but they lack the reasoning abilities to perform the task well.

H2. Fine-tuning affects both the representations and reasoning but the improvement in reasoning abilities primarily leads to better performance.

H3. Fine-tuning and adapters are not task-agnostic because the task-specific training hurts their ability to transfer reasoning.

We now analyze each of the task adaptation approaches through the lens of the above hypotheses. We perform all experiments with three different classes of language models (GPT-Neo, Pythia, Bloom) across a collection of 6 binary classification tasks. See Appendix B for further details.

**In-context learning: LLMs lack reasoning abilities.** We begin by studying the representation and reasoning gaps, as defined in eq. (1), for in-context learning. In Figure 3a, we plot the average accuracy across datasets for in-context learning, task-specific fine-tuning, and linear probing. We see that across models and different numbers of in-context examples, the reasoning gap $\Delta_{\text{reas}}$ accounts for up to 79.11% of the performance gap between in-context learning and fine-tuning. This indicates that the LLM representations have sufficient information but they lack the ability to reason over them.

**Fine-tuning: Improves task-specific reasoning.** We next investigate how fine-tuning for a specific task affects the performance of the base model. In Figure 3b, we show a scatter plot of the gains that can be attributed to improved representations against the reasoning gains. We see that, across models, reasoning improvements accounts for 73.06% of the improvements. This indicates that while fine-tuning improves both reasoning and representations of the LLM, the gains are predominantly due to improvements in task-specific reasoning. Furthermore, this task-specific fine-tuning of the LLM hurts its performance on other tasks. In Figure 3c, we show that the accuracy of a model fine-tuned on the AGNews dataset [48], leads to an average decrease of 25.77% on other tasks. Furthermore, this drop in accuracy can be attributed to the drop in task-specific reasoning capabilities—these account for 72.58% of the drop (see Appendix B for more details).

**Adapters: Impairs task-agnosticity via reasoning.** Task-specific adapters do not change the underlying representation ability of the model. To study their ability to generalize across tasks, we train an adapter for the AGNews dataset and evaluate it on other tasks. In Appendix B, we show that the performance drops across tasks by an average of 19.8%, indicating that adapters only learn task-specific reasoning abilities.

# 3   TART: Task-Agnostic Reasoning Transformers

The above analysis showed how it is the effective reasoning capabilities of the LLMs which limits its performance when compared with task-specific adaptation approaches. Building on this insight, we propose TART, which learns a general-purpose reasoning module completely agnostic to the underlying base LLM and when composed with any LLM via its embeddings, generically improves upon its reasoning abilities. TART is a completely task-agnostic method which works across a suite of tasks without any task-specific training.

## 3.1   Overview of algorithm

TART comprises two components: a generic task-agnostic reasoning module, and embeddings from the base LLM. The reasoning module is trained using only synthetic data (Gaussian logistic regression problems), agnostic of the auto-regressively trained language model, with the objective of learning to perform probabilistic inference (Section 3.2). This learned transformer module is then composed with the base LLM, without any training, by simply aggregating the output embedding and using those as an input along with the class label (Section 3.3). Together, these components make TART task-agnostic, boost performance quality by improving reasoning, and make the approach data-scalable by aggregating input embeddings into a single vector.

## 3.2   Reasoning module: Can transformers learn to do probabilistic inference?

TART's reasoning module is a Transformer-based model which is trained to perform probabilistic inference in-context using only synthetically generated data.

### 3.2.1   Training the reasoning module

The reasoning module is a Transformer model which is auto-regressively trained on a *family* of logistic regression tasks, with each input sequence corresponding to a different logistic regression problem. We next describe the model architecture and the training procedure.

**Model architecture.**   The reasoning module is based on the standard decoder-only Transformer architecture from the GPT-2 family. The GPT-2 architecture comprises a causal (unidirectional) transformer with 12 decoder layers and 8 attention heads per layer for a total of 22 M parameters. The model has absolute positional encodings for each position. Please see Appendix C.1 for details. The architecture takes as input a sequence of vectors and is trained to predict the next vector in the sequence. The input sequence consists of $k$ pairs of labeled examples $(x_1, y_1), (x_2, y_2), \ldots, (x_k, y_k)$, with each example $z_i = (x_i, y_i)$ using only two input positions of the transformer – one for the covariates $x$ and the other for the label $y$. This is in contrast to standard LLMs where each example is spread over multiple tokens which limits how many examples can be put in the context. For example, with a context window of 2048, our module can support 1024 examples while the base model can support only 10 examples, assuming each demonstration comprises 200 natural language tokens.

**Training procedure.**   This module is trained using gradient descent to minimize the population loss

$$\ell(T_\theta) := \mathbb{E}_{x,y} \left[ \frac{1}{k} \sum_{i=1}^{k} \ell_{\mathsf{CE}}(T_\theta(z_{1:i-1}, x_i), y_i) \right] , \tag{2}$$

where $z_{1:i-1}$ corresponds to the first $i-1$ examples and $\ell_{\mathsf{CE}}$ is the cross-entropy loss evaluated on the transformer prediction and the true $y_i$. Each training sequence $s_t$ used to update the parameters $\theta$ comprises a different $d$-dimensional logistic regression problem, sampled as

$$\text{Sequence } s_t : w_t \sim \mathcal{N}(0, I_d), \quad x_{i,t} \sim \mathcal{N}(0, I_d), \quad y_{i,t} \sim \sigma(\alpha \langle x_{i,t}, w_t \rangle) \qquad \text{for } i \in [k] , \tag{3}$$

where $\sigma$ represents the sigmoid function and the multiplier $\alpha$ determines the noise level of the problem. The above sampling model comprises three components, the regressor $w_t$, the covariates $x_{i,t}$ and the labels $y_{i,t}$ sampled as

- the regressor $w_t$ is sampled from a standard Normal distribution
- the covariates $x_{i,t}$ are sampled from a standard Normal distribution
- the corresponding $y_{i,t}$ are sampled based on the sigmoid of the inner product $\langle x_{i,t}, w_t \rangle$

See Figure 2 (Left) for a demonstration of a 2d dataset sampled from this process. The process is sampling noisy-linearly separable data in $d$ dimensions. We train our model with $d = 16$ and

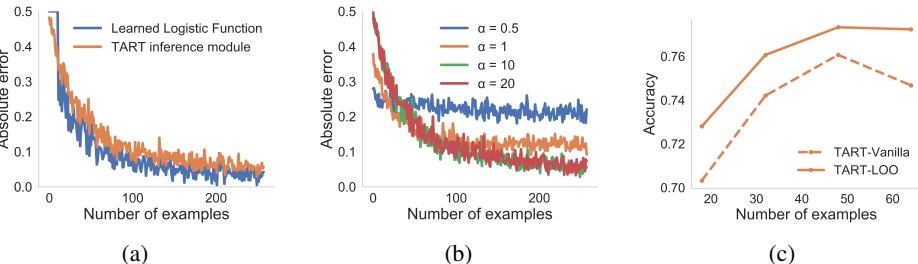

Figure 4: **Properties of TART's inference module**. (a) Comparison with learned logistic function: inference module recovers underlying probabilities. (b) Variation in error with different noise levels for model trained on $\alpha = 10$. (c) Comparison of TART performance when using LOO embeddings and vanilla embeddings.

$k = 256$. We describe the model hyper-parameters and the training procedure in more detail in Appendix C.1. Observe that the embedding dimension of the base LLM and the input dimension of the reasoning module might not always match. In order to compose these models together, we perform a dimensionality reduction via PCA to reduce the embedding dimension.

### 3.2.2 Properties of reasoning module

The task-agnostic reasoning module described above is trained to perform well on a family of logistic regression tasks. We study some properties of the reasoning module, in particular how well it learns to perform the task at an instance level and how robust is it to variations in the noise level $\alpha$.

**Accuracy of probabilistic inference.** For understanding the instance level performance of our reasoning module, we evaluate it on a sample of $64$ different logistic regression problems, sampled according to eq. (3). For each problem, we train task-specific linear classifiers using logistic regression and compare them with our task-agnostic reasoning module. In Figure 4a we plot the deviation of the predicted probabilities (averaged over the $64$ problems) from the true probabilities for our reasoning module and the task-specific logistic solvers as a function of the number of examples used for predictions. We observe that the error for our reasoning module decreases as a function of the number of in-context examples and is within $2\%$ of the task-specific logistic function.

**Robustness to noise level.** We study the robustness of the learned module to the noise levels, $\alpha$, of the logistic regression problem. Recall that we trained our inference module by fixing the noise level $\alpha = 10$. At inference time, we vary the noise level to $[0.5, 1, 10, 20]$, where lower values corresponds to noisier problem. The reasoning module generalizes to easier problem without any drop in accuracy but as we make the problem harder ($\alpha = [0.5, 1]$), the error increases progressively (see Figure 4b).

### 3.3 Role of representations: How to embed training examples?

The reasoning module composes with a base LLM through its final layer embeddings. A natural way to produce these embeddings is to place all the train examples in-context and then average the embedding vectors corresponding to the particular example (see Figure 2). At inference time, we append the test example to the training set, and average the embeddings corresponding to this example. We call these vanilla embeddings. Our experiments reveal that these embeddings seem to saturate (or even hurt performance) beyond a certain number of in-context examples (see Figure 4c). One reason can be that the causal nature of the model causes these embeddings to have asymmetric information—the embeddings of each example is influenced by its preceding examples.

To counter this asymmetry, we propose *leave-one-out* (LOO) embeddings where the embeddings for each training point is formed by placing all the other train examples before it in the prompt such that all the embedding are formed with the same information content. In Figure 4c, changing the embedding style from vanilla to LOO consistently improves performance across models and tasks. The LOO-embeddings help TART be data-scalable by enabling it to embed a much larger number of points than the context window can support. To do so, we use only a subset of the train examples as the in-context prompt. The reasoning module, by its architecture design, can already accommodate many more examples than supported by the context window of the base LLM.

### 3.4 Theoretical analysis: Generalization of TART to natural language tasks

We study the generalization properties of the proposed task-agnostic method TART. Note that that the inference module is trained completely on synthetic data while at evaluation time, our input is the embeddings from a natural language task. In Theorem 1 we show that its performance on the natural language task depends on the distribution shift from the synthetic to the true distribution (see Appendix C.3 for a formal statement and proof).

**Theorem 1** (Informal). *Let $\mathcal{T}$ represent the class of transformer models and $T_S \in \mathcal{T}$ denote the trained reasoning module on set $S$ of synthetic regression with $n_{syn}$ sequences sampled from distribution $P_{syn}$ in eq. (3). The error of the transformer $T_S$ when evaluated on a distribution $P_{NL}$ over natural language sequences is*

$$\text{err}_{P_{NL}} \lesssim W_1(P_{NL}, P_{syn}) + \sqrt{\frac{Comp(\mathcal{T})}{n_{syn}}} + \hat{\text{err}}_{P_{syn}}(T_S) \,, \tag{4}$$

*where $W_1$ denotes the Wasserstein-1 metric, $Comp(\mathcal{T})$ represents the complexity of class $\mathcal{T}$, and $\hat{\text{err}}$ represents the error on the empirical distribution.*

A few comments are in order: The first term represents the distribution shift error between the true natural language task and the synthetic task. The second term corresponds to the generalization error on the logistic regression task, which can be made arbitrarily small because scales with $n_{syn}$, the number of synthetic datapoints which can be generated without any cost. The third term above is the optimization error indicating how well has the reasoning module $T_S$ fit to the synthetic training set.

Observe that for our algorithm TART, the distribution $P_{NL}$ corresponds to the distribution induced on the embeddings of the natural language data (post the PCA and renormalization) while the distribution $P_{syn}$ is the one corresponding to the sampling process in equation (3).

## 4 Experimental evaluation

We evaluate TART on a wide range of binary classification tasks across three domains: language, vision and audio. We demonstrate that TART improves base in-context performance and closes the gap with standard task-specific strategies. We also conduct ablations to demonstrate that TART scales with model size and can support 10x more samples than in-context learning.

### 4.1 Experimental setup

**Datasets.** We briefly describe the datasets used, with details available in Appendix E.1. We consider 14 different binary classification tasks ranging from sentiment classification, news article categorization to spam detection. The evaluation datasets include: SST [35], Rotten Tomatoes [29], SMS Spam [3], IMDB [27], Civil Comments [8], AGNews [48], DBPedia [48], and the Youtube dataset [46]. Since AGNews and DBPedia14 are multi-class datasets, we construct 4 binary classification tasks from each dataset respectively. For each dataset, we truncate the input text to be at most 100 characters to enable us to fit sufficient number of samples in-context.

**Model families.** We evaluate our method across three different families of models: GPT-NEO [6], PYTHIA [5], and BLOOM [33]. For our evaluations across 14 datasets, we use GPT-NEO (125M), PYTHIA (160M) and BLOOM (560M). For ablations on larger models, we evaluate models with 1B parameters across each of the model families (i.e., GPT-NEO (1.3B), PYTHIA (1.4B) and BLOOM (1.7B)) and models with 3B parameters (i.e., GPT-NEO (2.7B), PYTHIA (2.8B) and BLOOM (3B)). We additionally evaluate on GPT-J (6B) [37].

**Baselines.** We evaluate our models against all types of task-adaptation strategies described in Section 2.2: 1) in-context learning, 2) full fine-tuning, 3) last layer fine-tuning, 4) LM head fine-tuning, and 5) adapters. For each baseline, we perform an extensive hyper-parameter search over number of epochs and learning rate for each dataset in order to optimize performance (see Appendix E.1 for hyperparameter details). For TART, we chose a base default set of parameters and use the *same* inference module with the exact same weights for all the experiments in this section.

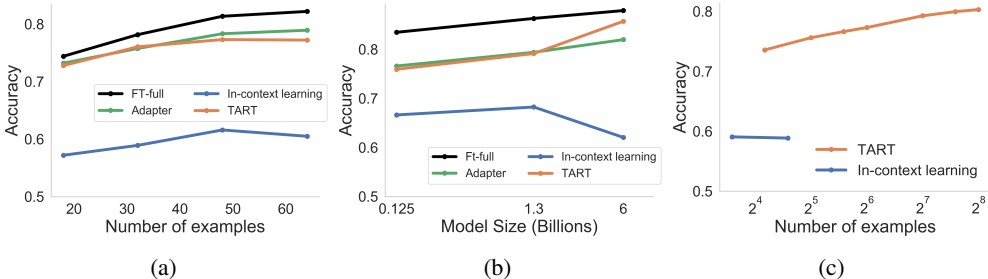

Figure 5: **Effects of scale**. (a) Effect of number of in-context examples on performance for different task adaptation strategies. (b) Effect of model size on the performance of different task adaptation strategies. (c) Beyond context length limitations, performance comparison with respect to number of in-context examples.

## 4.2 Natual language benchmark evaluations

**Performance with respect to baselines.** As shown in Appendix E.2, averaged across all tasks and model families, TART improves upon the base in-context learning performance by an average of $18.4$ points, improves upon adapter heads by $3.4$ points, and is within $3.1$ points of full fine-tuning. We also observe that TART consistently outperforms the task specific strategies of LM head fine-tuning and last layer fine-tuning.

**Performance on RAFT Benchmark** We evaluate TART on all binary classification tasks in the RAFT Benchmark, following the protocol used in HELM [22]. When applied with GPT-NEO (125M), TART [0.634] outperforms BLOOM (176B) [0.595], and is within $4\%$ points of GPT-3 (175B) [0.673], both of which are 1000x larger in size.

**Performance with number of in-context examples.** Our results demonstrate that performance of TART scales with number of in-context examples (see Figure 5a). Across 14 tasks and 3 model families, when scaling from 18 to 64 examples, TART improves performance by an average of $4.8\%$. Correspondingly, full fine-tuning improves performance by $9.0\%$.

**Scaling with base model size.** We analyze how different task-adaptation strategies scale with respect to model size using the GPT-Neo family: GPT-NEO (125M), GPT-NEO (1.3B) and GPT-J (6B). Figure 5b shows that when scaling from 100M to 6B parameters, performance of task-specific methods and TART increases as a function scale. For TART, the performance increases by $9.8\%$ while using the same inference module across model sizes. Furthermore, the difference in performance between TART and fine-tuning baseline reduces from $7.5\%$ to $2.2\%$ from the 100M scale to 6B scale.

**Beyond context length.** We evaluate the data-scaling properties for both in-context learning and TART (Figure 5c). To demonstrate the scaling property, we do not truncate the input text to 100 characters and utilize the entire text sequences. For TART, we observe that accuracy continues to improve when scaling from 18 to 256 in-context examples with $6.8\%$ lift in performance. In comparison, ICL, which is bottlenecked by context length, supports 10x less samples, with the context window saturating at 24 examples only and lags TART by an average of $19.1\%$.

**Extension to other architectures.** We compare the performance of TART by changing the underlying architecture used to train the reasoning module. Specifically, we train a reasoning module that has the Hyena architecture [30] using the same synthetic logistic regression tasks. In appendix E.3, we show that the accuracies obtained by swapping the Transformer with the Hyena module maintains the same accuracy as before. Thus, TART's performance is robust to the underlying architecture.

**Performance in small sample regime.** While our focus has been on scaling up number of in-context examples (ranging from 18-64), in Appendix E.4 we compare the performance of TART with fine-tuning and in-context learning in the small sample regime. Specifically, we look at few-shot learning with number of examples varying from 4-10. Across 6 datasets, TART outperforms fine-tuning by $4.1\%$ (with max. gains up to $20.6\%$) and in-context learning by $11.6\%$ (with max. gains up to $35.7\%$). See Figure 26 for detailed results.

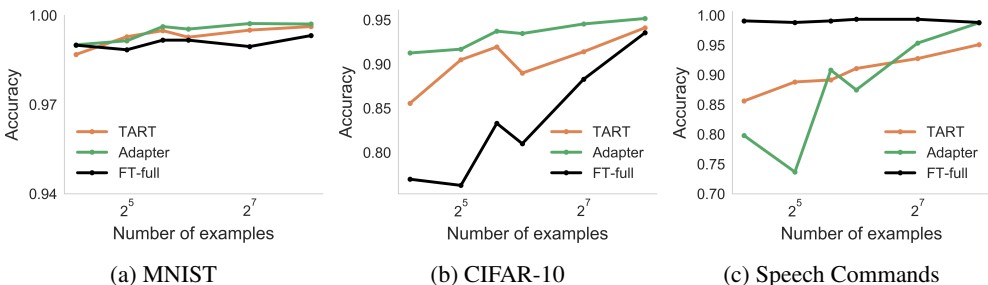

(a) MNIST        (b) CIFAR-10        (c) Speech Commands

Figure 6: TART can generalize across domains using the same inference module that was used for language benchmarks: Performance across vision tasks (MNIST, CIFAR-10) and an audio task (Speech Commands).

### 4.3 Extensions to other modalities

We demonstrate that TART is not only agnostic to models and tasks, but also modalities. We extend TART to classification tasks on modalities beyond language: vision and audio. For vision tasks, we use representations from Google's 307M parameter pretrained Vision Transformer (ViT) model [44]: VIT-LARGE-PATCH16-224. For audio tasks, we use representations from OpenAI's 1.5B parameter pretrained Whisper model [31]: WHISPER-LARGE. In applying TART to the representations from these models, we provide a way for performing in-context learning in modalities beyond text. We refer the reader to Appendix E.5 for further details on the experiment setup.

**Vision application.** We evaluate the performance of TART on binary classification versions of MNIST [19] (classes 0 and 8) and CIFAR-10 [17] (classes plane and bird). As shown in Figure 6a and 6b, performance of TART is competitive with task-specific adaptation approaches.

**Audio application.** We evaluate TART on a binary classification version of the Speech Commands dataset [40], where the task is to classify "stop" and "go" utterances. As shown in Figure 6c, performance of TART is competitive with task-adaptation approaches.

### 4.4 Extension to multi-class classification problems

While we focused on binary classification tasks in this paper, in Appendix D we demonstrate how to extend TART to multi-class problems. Instead of training a module on synthetic binary classification problem, we modify the synthetics to sample from a multinomial extension of the logistic model. We evaluated this multi-class model on 4-class classification tasks (AGNews, Ecommerce) as well as a 14-class classification problem (DBPedia-14). Our evaluations show that TART can outperform in-context learning by up to 45.9% on AGNews, 52.6% on Ecommerce and 59.3% on DBPedia-14.

## 5 Discussion

We look at the problem of task-agnostic learning with LLMs. We show that LLMs lack the ability to perform simple reasoning over their learned representations and introduce TART, a task, model and domain agnostic method for improving their reasoning abilities. In this work, we focus on classification tasks, showing that synthetic, logistic regression task data can be used to train a generic reasoning module capable of completing this class of tasks. In future work, we seek to understand whether synthetic tasks exist for training other generic reasoning modules, capable of improving base LLM performance on tasks such as generation or summarization.

### Acknowledgements

We are grateful to Simran Arora, Rishi Bommasani, Niladri Chatterji, Arjun Desai, Sabri Eyuboglu, Neha Gupta, Karan Goel, Erik Jones, Ananya Kumar, Cassidy Laidlaw, Megan Leszczynski, Piero Molino, Laurel Orr, Michael Poli, Dimitris Tsipras, Michael Wornow, Ce Zhang, and Michael Zhang for their helpful comments and feedback, and discussions which helped shape this project. We thank Neel Guha for helpful discussions in shaping the narrative of this paper.

We gratefully acknowledge the support of NIH under No. U54EB020405 (Mobilize), NSF under Nos. CCF1763315 (Beyond Sparsity), CCF1563078 (Volume to Velocity), and 1937301 (RTML);

US DEVCOM ARL under No. W911NF-21-2-0251 (Interactive Human-AI Teaming); ONR under No. N000141712266 (Unifying Weak Supervision); ONR N00014-20-1-2480: Understanding and Applying Non-Euclidean Geometry in Machine Learning; N000142012275 (NEPTUNE); NXP, Xilinx, LETI-CEA, Intel, IBM, Microsoft, NEC, Toshiba, TSMC, ARM, Hitachi, BASF, Accenture, Ericsson, Qualcomm, Analog Devices, Google Cloud, Salesforce, Total, the HAI-GCP Cloud Credits for Research program, the Stanford Data Science Initiative (SDSI), and members of the Stanford DAWN project: Facebook, Google, and VMWare. CDS was supported by a NSF CAREER (award 2046760).

The U.S. Government is authorized to reproduce and distribute reprints for Governmental purposes notwithstanding any copyright notation thereon. Any opinions, findings, and conclusions or recommendations expressed in this material are those of the authors and do not necessarily reflect the views, policies, or endorsements, either expressed or implied, of NIH, ONR, or the U.S. Government.

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

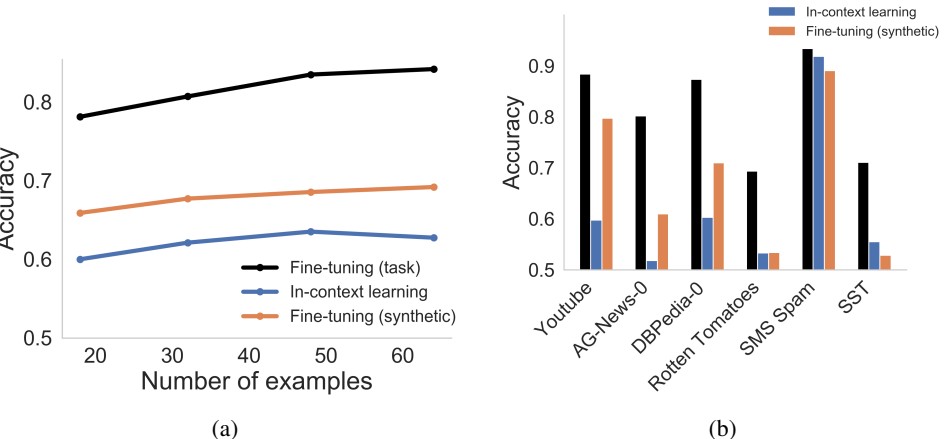

(a)                                          (b)

Figure 7: **Fine-tuning with NL synthetic task**. (Left) Averaged over 6 different tasks, fine-tuning with the NL synthetic task provides a lift over base in-context learning, and scales with number of examples. (Right) Dataset level comparisons between task-specific fine-tuning, in-context learning and synthetic fine-tuning: synthetic fine-tuning outperforms base in-context learning on 4 out of 6 datasets, but lags task-specific tuning.

## A  Fine-tuning model with NL-based Probabilistic Inference Tasks

As highlighted in Section 1, we describe the details for *directly* fine-tuning an LLM on synthetically generated probabilistic inference tasks to improve reasoning capabilities. For the following experiments, we use GPT-NEO (125M) as the base model.

### A.1  Training Task

We fine-tune the base model using a sequence of $k$ pairs of synthetically generated labeled natural language examples $(x, y)$. Each example $x$ in the sequence $s = (x_1, y_1), \ldots, (x_k, y_k)$ consists of a list of strings constructed from a fixed $V$ size of dimension $d = 30$ . We use the following fixed vocabulary: [ "sports", "love", "hate", "car", "school", "family", "work", "sleep", "water", "tree", "fox", "train", "random", "movie", "music", "book", "play", "house", "spell", "bar", "jump", "park", "run", "hill", "fast", "slow", "talk", "wallet", "orange", "apple", "ball", "cat" ].

To generate a particular example $x_i$, we sample each coordinate $x_{i,j}$ uniformly from the set $\{-1, +1\}$. If the sampled value is $+1$, we set the value to be the corresponding word in the vocabulary, that is, $x_{i,j} = V_j$. Otherwise, the word $x_{i,j}$ is set to "null". For a given sequence $s$, we generate each of the labels $\{y_i\}$ as:

$$w_t \sim \mathcal{N}(0, I_d), \quad y_i \sim \sigma(\alpha \langle x_i, w \rangle), \text{for } i \in [k] , \tag{5}$$

where we set noise parameter $\alpha = 5$. If the sampled output is 0, we set the $y_i$ to "negative" and "positive" otherwise.

Finally, the inputs are formatted with following template: "$x_1 : y_1 , x_2 : y_2 , \ldots , x_k : y_k$" and the model is trained using gradient descent on the loss

$$\ell(T_\theta) := \mathbb{E}_{x,y} \left[ \frac{1}{k} \sum_{i=1}^{k} \ell_{\mathsf{CE}}(T_\theta(z_{1:i-1}, x_i), y_i) \right] , \tag{6}$$

where $z_{1:i-1}$ corresponds to the first $i-1$ examples and $\ell_{\mathsf{CE}}$ is the cross-entropy loss evaluated on the transformer prediction and the true $y_i$.

More concretely, a sample input sample sequence $s$ to be used for training looks like:

```
"sports love null car ... cat: positive,
null love null car ... null: negative,
...
sports null hat null ... cat : positive"
```

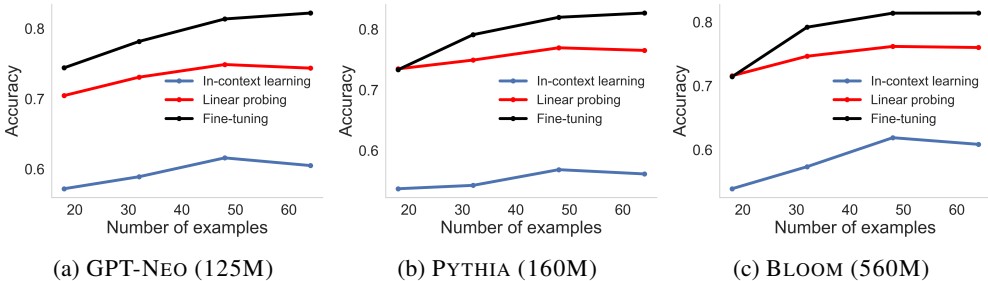

Figure 8: **Comparison of linear probing, in-context learning, and fine-tuning**. Accuracy of in-context learning vs. linear probing on model embeddings across three model families: representations have sufficient information.

## A.2 Training Parameters

We train GPT-NEO (125M) on this synthetic task with a learning rate of 0.0001 and a batch size of 4. For each sequence we sampled a total of $k = 60$ examples and trained the model for 10000 steps.

## A.3 Evaluation

We evaluate on 6 datasets: AG News [48], DBPedia [48], SST [35], SMS Spam [3], Youtube [46] and Rotten Tomatoes [29]. We truncate the input texts to 100 characters to fit more in-context examples. We evaluate over a range of context sizes ($k$=[18, 32, 48, 60]). At evaluation time, we use the same "sentence : label" format that was used to train the model. We evaluate over 3 random seeds. In Figure 7, we compare the performance of the model fine-tuned on probabilistic inference tasks and the base in-context learning performance. While the performance of the fine-tuned model is better than the base in-context learning capabilities, task-specific fine-tuning still outperforms it by an average of 16.87% (see Figure 7).

## B  Details for Representation-Reasoning decomposition evaluations

In this section, we provide details for the experimental evaluation and additional results for the representation-reasoning decomposition introduced in Section 2.3.

### B.1  Experimental setup

For these experiments, we evaluate three different language models: GPT-NEO (125M), PYTHIA (160M), and BLOOM (560M) on a collection of 6 binary classification datasets: AG News [48], DBPedia [48], SST [35], SMS Spam [3], Youtube [46] and Rotten Tomatoes [29]. For each model, we run evaluations for three different random seeds, where the randomness was in the set of datapoints chosen for the training task. For the hyperparameters, we performed an extensive search for all models across datasets. For details on these hyperparameters and the adapter architecture we evaluate over, see Appendix E.1.

To conduct linear probing over the embeddings, we perform logistic regression over the output embeddings of each model and the given labels in the training set using the built-in logistic regression solver from the scikit-learn python library, utilizing the *lbgfs* solver.

### B.2  Detailed results

For each class of methods in the task-adaptation taxonomy from Section 2.2, we now describe the details of the experimental evaluation and present additional results.

**In-context learning.**    To understand the representation and reasoning gaps for in-context learning, we evaluated three accuracies: a) using in-context learning with base models, b) fine-tuning the model for the task, and c) linear probing the model specifically for the task. The gap due to representation

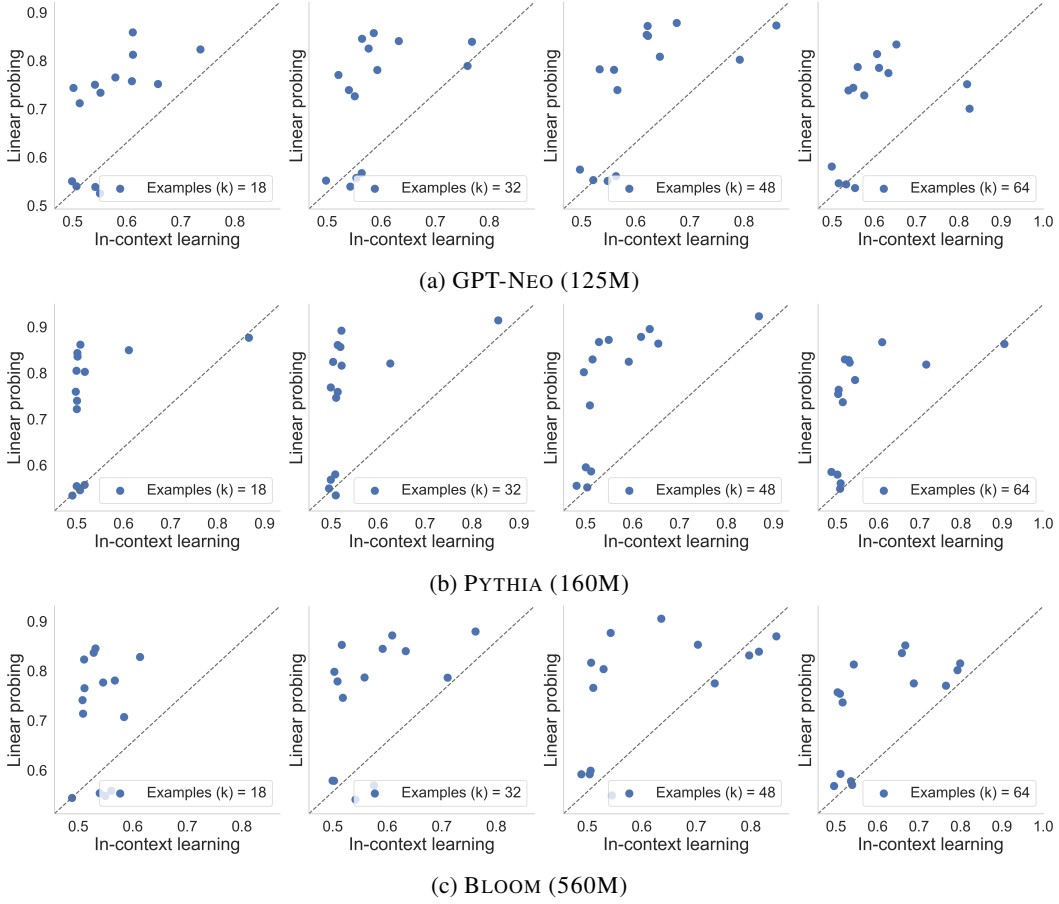

Figure 9: **Linear probing vs. in-context learning**. Scatter plot of accuracy of in-context learning vs. linear probing on model embeddings across model families and different number of in-context examples: linear probing consistently outperforms in-context learning indicating that the learned representations have sufficient information. Each point in the plot represents a dataset.

was taken to be the difference between the fine-tuning and linear probing accuracies while the reasoning gap was the gap between linear probing and in-context learning, as described in eq. (1).

In Figure 8, we show the average accuracies of in-context learning, linear probing, and fine-tuning across the 6 tasks. Linear probing closes the gap between in-context learning and fine-tuning, while being task-specific. In Figure 9, we show a scatter plot of the accuracies of linear probing vs. the accuracies of base in-context learning. Linear probing consistently out performs in-context learning showing that the learned representations across these models have sufficient information to complete the tasks but lack reasoning abilities.

**Fine-tuning.** For the fine-tuning approach, we are interested in understanding two hypotheses: a) how does fine-tuning improve the model performance, and b) whether fine-tuning hurts task-agnosticity of the base model and if yes, what is the underlying reason for it.

For the first hypothesis, we evaluate the proportion of gains that can be attributed to improved representations of the the underlying model. This is computed as the difference in performance of linear probing over the base model and over the fine-tuned model — this evaluates how much the representations have changed specifically for this task. The reasoning gains are then computed by subtracting the representation gains from the total gain (fine-tuning accuracy minus in-context accuracy). Figure 10 shows a scatter plot of these representation gains and reasoning gains, plotted across different datasets and number of examples ($k$). Most of the gains which are realized by fine-tuning are because of improved task-specific reasoning capabilities across the model families.

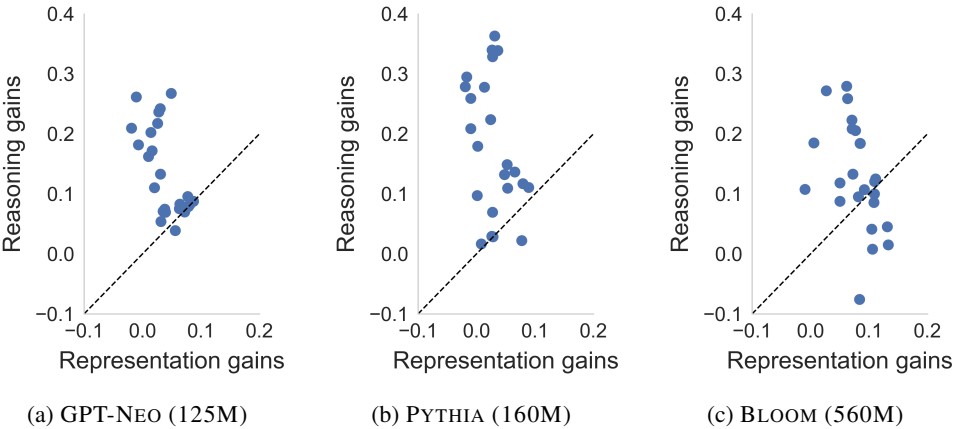

(a) GPT-NEO (125M)      (b) PYTHIA (160M)      (c) BLOOM (560M)

Figure 10: **Effects of fine-tuning on reasoning**. Across datasets (each point in plot represents a dataset) and model families, fine-tuning improves task-specific reasoning which improves it performance over base in-context learning.

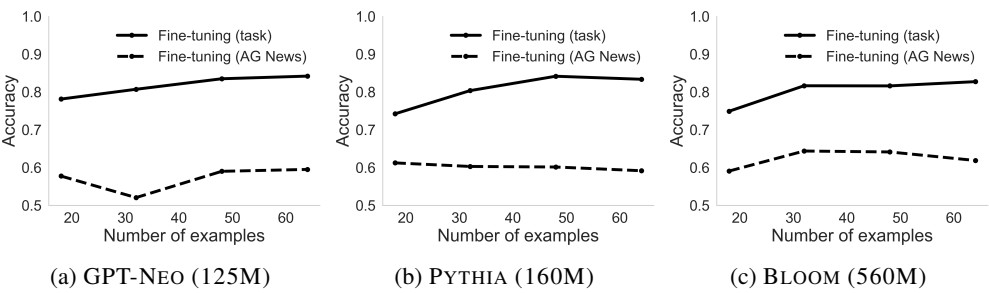

(a) GPT-NEO (125M)      (b) PYTHIA (160M)      (c) BLOOM (560M)

Figure 11: **Effects of fine-tuning on task-agnosticity** Accuracy of task-specific fine-tuned model vs. accuracy of model fine-tuned on AG-News-0 and evaluated on task. Fine-tuning hurts task-agnosticity across all three model families.

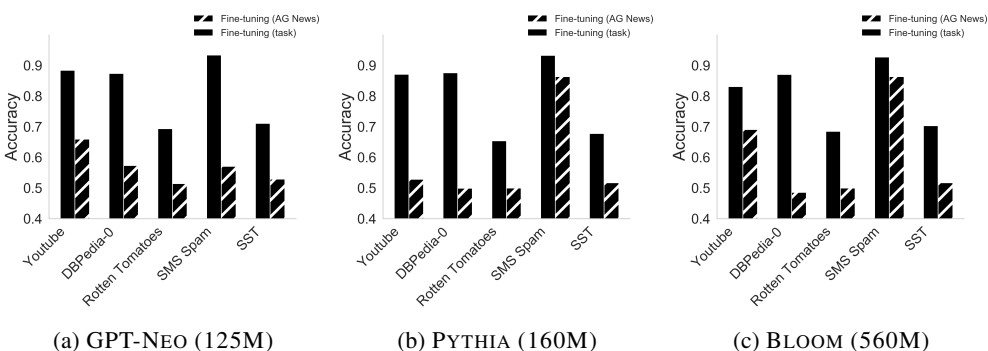

(a) GPT-NEO (125M)      (b) PYTHIA (160M)      (c) BLOOM (560M)

Figure 12: **Effects of fine-tuning on task-agnosticity (dataset level)** Accuracy of task-specific fine-tuned model vs. accuracy of model fine-tuned on AG-News-0 and evaluated on task. Fine-tuning consistently hurts task-agnosticity across all three model families and datasets.

For the second hypothesis, we first evaluate *whether* fine-tuning hurts task-agnosticity. For this we evaluate two sets of accuracies: accuracy of a model fine-tuned for the specific task and the accuracy of a model on the task but fine-tuned on the AG News dataset. From Figures 11 and 12, we see that there is a drop in accuracy—over $25.77\%$ across models and datasets. For the second part, we again decompose the drop in accuracy into a representation drop and a reasoning drop. The representation drop is computed by training a linear probe over the two models (task-specific fine-tuned and AG News fine-tuned) and looking at the difference between them. The reasoning drop, as before, is computed by subtracting this representation drop from the total drop. Figure 13 shows that most of

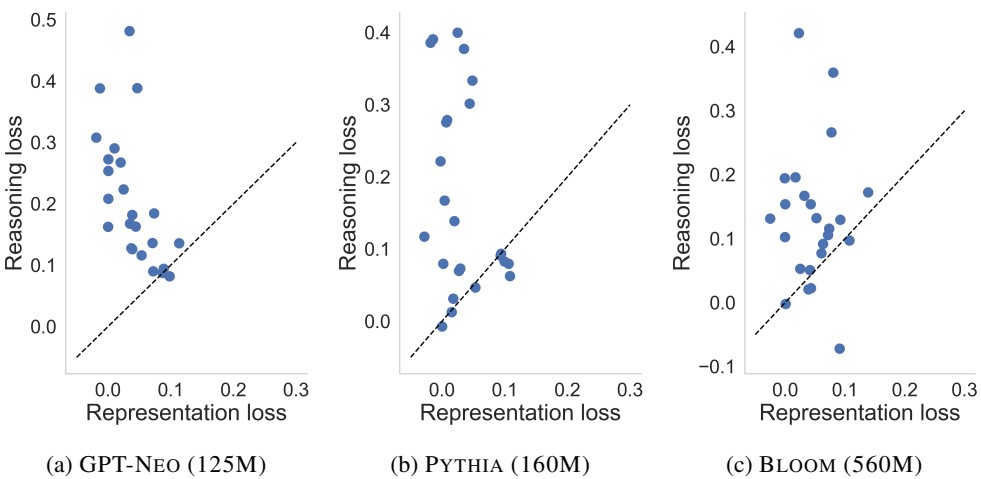

(a) GPT-NEO (125M)    (b) PYTHIA (160M)    (c) BLOOM (560M)

Figure 13: **Effects of fine-tuning on task agnosticity**. Scatter plot of reasoning loss against representation loss when the base model is trained on AG-News-0 and evaluated on other tasks. Across datasets (each point in plot represents a dataset), fine-tuning majorly impairs reasoning when transferring to tasks outside the specific fine-tuned task.

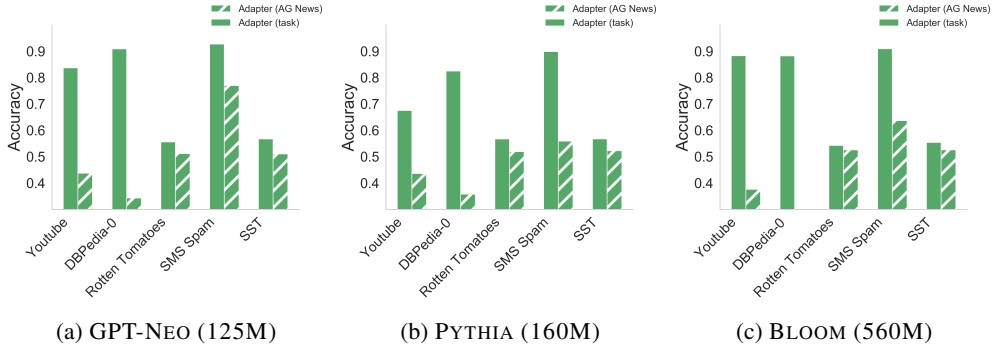

(a) GPT-NEO (125M)    (b) PYTHIA (160M)    (c) BLOOM (560M)

Figure 14: **Effects of fine-tuning on task-agnosticity of adapters**. Accuracy of task-specific fine-tuned adapter vs. accuracy of adapter fine-tuned on AG-News-0 and evaluated on task. Fine-tuning consistently hurts the generalization ability of adapters across datasets.

this drop in task-agnosticity can be attributed to over-fitting of the reasoning abilities over the task for which the models are fine-tuned.

**Adapters.** Since adapters do not modify the underlying representations of the model, we look at how a single adapter generalizes across tasks. For this we train an adapter on the AG News dataset and evaluate it on the other datasets. We compare this set of accuracies with those obtained by task-specific adapters in Figure 14. The main conclusion is that task-specific adapters are not agnostic learners and over-fit to the task for which they are fine-tuned.

## C    Details for TART implementation

This section contains the details on training TART's reasoning module and extended results on the choice of embeddings from Section 3.

### C.1    TART's reasoning module

**Architecture details.** We use the standard GPT-2 architecture [32] for training our reasoning module. We set the embedding size to 256, number of decoder layers to 12, and number of heads to 8 for a total of 22 million parameters. Since the GPT-2 backbone outputs a sequence of embeddings,

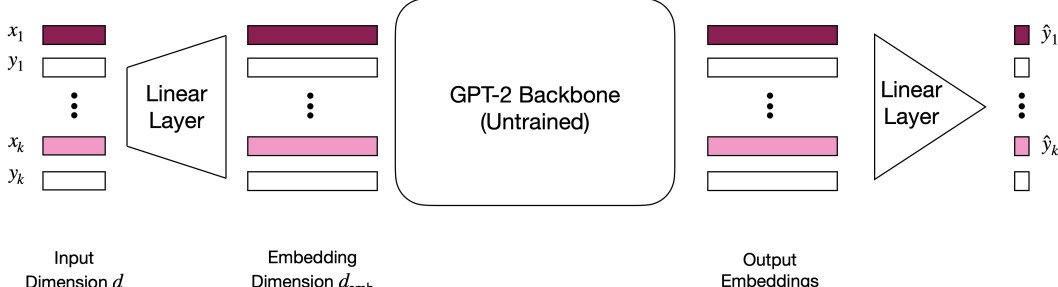

Figure 15: **TART reasoning module architecture**. The reasoning module takes as input sequences of $(x, y)$ pairs of dimension $d$. A linear layer is used to project $d$ to the hidden dimension size of the GPT-2 backbone. Finally, a linear layer is applied to the outputs of the backbone to generate predictions for each $x_k$ in the input sequence.

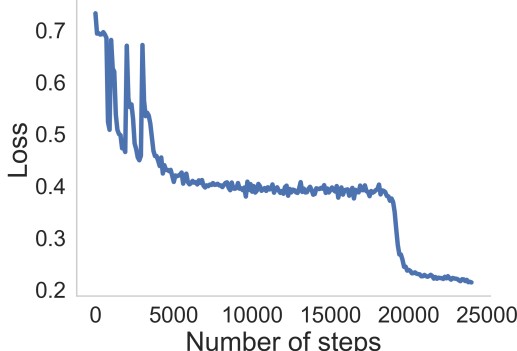

Figure 16: **Training loss vs. number of steps.** The plot shows the variation in training loss as a function of the number of steps of gradient descent for TART's reasoning module.

we additionally add a linear layer in the end to convert the output to scalar values (see Figure 15). Additionally, the binary labels $y$ are encoded as a one-hot vector to match the input dimension $d$ of the corresponding covariates $x$.

**Training procedure.** We trained TART's reasoning module with a context length of 258 (allowing for up to 129 in-context examples). The batch size was set to 64, learning rate to 0.0001 and the model was trained for a total of 24000 epochs. Each batch of training data consists of sampling a sequence of 258 examples using eq. (3). In addition to these hyperparameters, we used a curriculum on the input dimensions and on the number of examples in the sequence to train our module—the input dimensions started from a value of 4 and were incremented by 4 every 1000 epochs while the number of examples started from 18 and were incremented by 30 every 1000 epochs.

**Combining reasoning module with base LLM.** We trained the reasoning module with input dimension set to 16. However, most base models produce representations which are much higher dimensional (ranging from 784 to 2048). In order to reduce the dimensionality of these representations, we perform PCA on the output embeddings of the base model, learning the components using only the training points available for that specific task. The test examples are then projected onto these principal components to produce 16 dimensional input representations.

## C.2    Choice of representations

As discussed in Section 3.3 there are two possible options for forming the representations, the vanilla embeddings and the leave-one-out (LOO) embeddings. Figure 18 shows the schematic differences between the two style of embedding. In Figure 17, we plot the average accuracies across different datasets for both vanilla and LOO embeddings, observing that the LOO embeddings consistently perform better across the different model families.

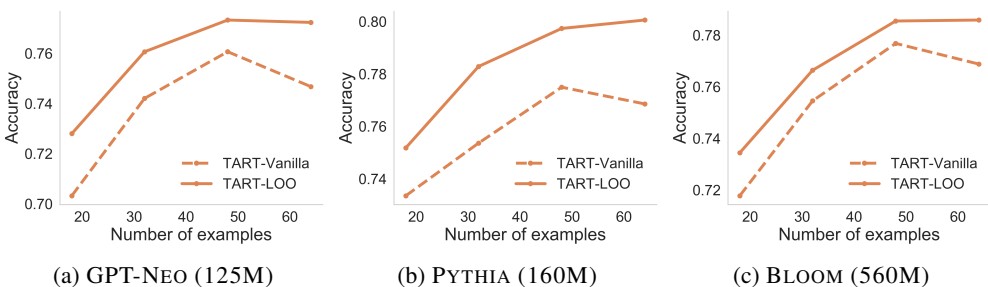

(a) GPT-Neo (125M)    (b) Pythia (160M)    (c) Bloom (560M)

Figure 17: **LOO embeddings vs. Vanilla Embeddings**. Comparison of Tart performance when using LOO embeddings and vanilla embeddings. Vanilla embeddings see a performance collapse, but LOO embeddings do not.

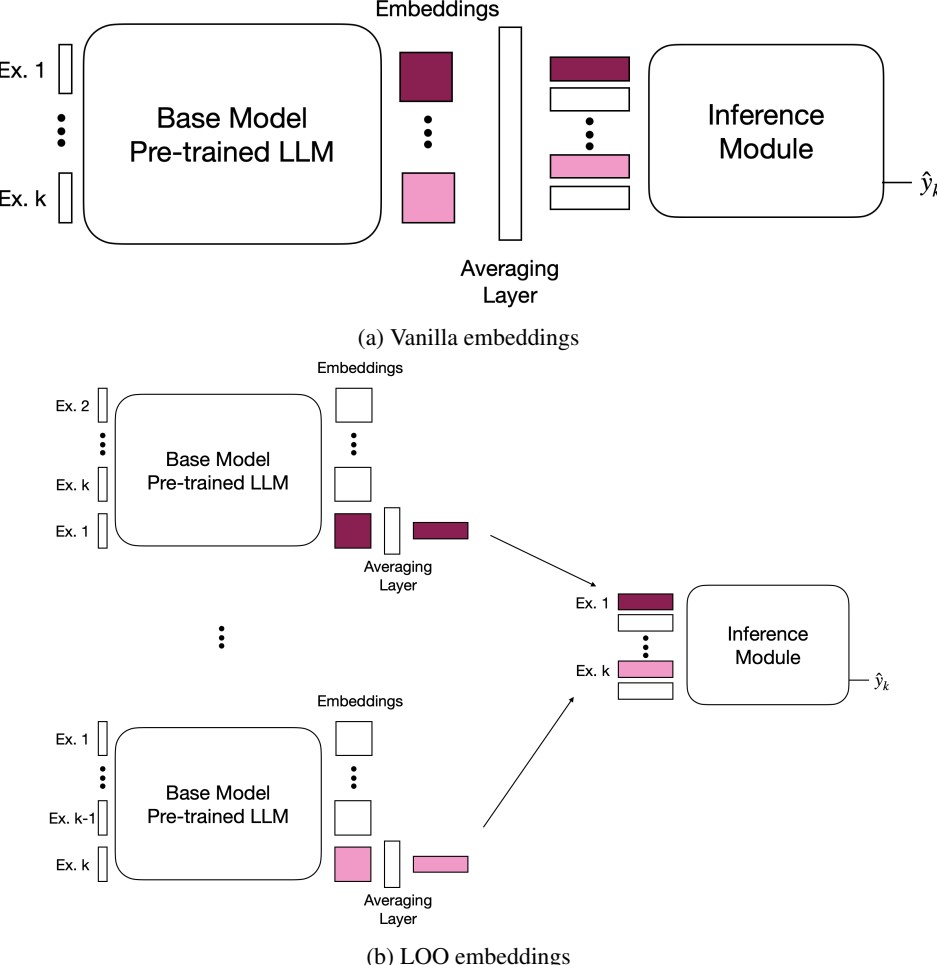

(a) Vanilla embeddings

(b) LOO embeddings

Figure 18: **Tart Embedding Protocols**. (a) For the vanilla embeddings, the test example is appended to the training set and the sequence is passed to the base model. The representation for each train example in this sequence is taken as the average embedding across all its tokens. (b) For the LOO embeddings, we generate embeddings for each train example separately by placing all the other train examples before it in the prompt and averaging the embeddings over the final example's tokens.

## C.3 Proof of Theorem 1

In this section, we provide a formal statement of Theorem 1 from Section 3.4. Our theorem quantifies the expected error of the Transformer, trained on synthetic data, on natural language tasks in terms of the change in the two input distributions.

We begin by introducing some notation. We denote the class of Transformer family by

$$\mathcal{T}_\Theta := \{T_\theta : \mathbb{R}^{(2k+1) \times d} \mapsto \mathbb{R} \quad | \quad \theta \in \Theta\}, \tag{7}$$

where $k$ represents the maximum number of in-context examples the Transformer can support, $d$ represents the input dimensions, and $\Theta$ represents the corresponding parameter class over which the Transformer family is defined.

Observe that the Transformer family $\mathcal{T}_\Theta$ takes as input a sequence of $k$ train examples, each corresponding to two tokens of hidden dimension $d$: a covariate $x \in \mathbb{R}^d$ and a binary label, encoded as a one-hot vector in $d$ dimension. This sequence of train examples is followed by a test example, for which we only have the features $x_{k+1}$.

Given this background, let $P_{\mathsf{syn}}$ denote the synthetic distribution over sequences $\{(x_1, y_1), \ldots, (x_k, y_x), (x_{k+1})\}$. Similarly, let $P_{\mathsf{NL}}$ denote the corresponding distribution over sequences derived from natural language tasks where $x_i$ denotes the LLM embeddings of the example. Recall from Section 3.2.1, the synthetic training distribution $P_{\mathsf{syn}}$ is given by

$$\text{Sequence } s_t : w_t \sim \mathcal{N}(0, I_d), \quad x_{i,t} \sim \mathcal{N}(0, I_d), \quad y_{i,t} \sim \sigma(\alpha \langle x_{i,t}, w_t \rangle) \qquad \text{for } i \in [k], \tag{8}$$

for each training point $(x_{i,t}, y_{i,t})$. The test point is also sampled similarly from an independent standard normal distribution. Let $\ell : \mathbb{R} \times \mathbb{R} \mapsto \mathbb{R}$ be the loss function used for evaluating the performance of the reasoning module. Further, let use denote the expected loss under a distribution $P$

$$\mathsf{err}_P(T) := \mathbb{E}_{(s,y) \sim P}[\ell(T(s), y)], \tag{9}$$

and the corresponding empirical distribution over samples $S$ by $\hat{\mathsf{err}}_P$, where the dependence on the samples is implicit. Given these samples, we denote the empirical risk minimizer

$$T_S = \arg \min_{T \in \mathcal{T}_\Theta} \frac{1}{|S|} \sum_{s \in S} \ell(T(s), y). \tag{10}$$

In addition to these notation, we make the following Lipschitz assumption on the the loss function $\ell$ and the Transformer model $T$.

**Assumption 1.** *[Lipschitz loss.] For any two output labels $y, y'$, the loss function $\ell$ is Lipschitz with constant $c_1$, that is,*

$$|\ell(Tr(s), y) - \ell(Tr(s), y')| \leq c_1 |y - y'|. \tag{11}$$

**Assumption 2.** *[Lipschitz models.] For any two input sequences $s, s'$, each any model $T \in \mathcal{T}_\Theta$ is Lipschitz with constant L, that is,*

$$|T(s) - T(s')| \leq L \|s - s'\|. \tag{12}$$

Given this setup, we are now ready to state a formal version of Theorem 1.

**Theorem 2** (Formal version of Theorem 1)**.** *Let $T_S \in \mathcal{T}_\Theta$ denote the trained reasoning module on set $S$ of synthetic logistic regression tasks with $n_{\mathsf{syn}}$ sequences sampled from distribution $P_{\mathsf{syn}}$ in eq. (3). Let the loss function $\ell$ satisfy Assumption 1 and the model class $\mathcal{T}_\Theta$ satisfy Assumption 2. Then, with probability at least $1 - \delta$, we have*

$$\mathsf{err}_{P_{\mathsf{NL}}}(T_S) \leq c_1 \max(1, L) \cdot W_1(P_{\mathsf{NL}}, P_{\mathsf{syn}}) + c_1 \cdot \sqrt{\frac{2VC(\mathcal{T}_\Theta) \ln m}{n_{\mathsf{syn}}}} + 4\sqrt{\frac{2 \ln(4/\delta)}{n_{\mathsf{syn}}}} + \hat{\mathsf{err}}_{P_{\mathsf{syn}}}(T_S), \tag{13}$$

*where $W_1$ denotes the Wasserstein-1 metric and $VC(\mathcal{T}_\Theta)$ represents the VC dimension of class $\mathcal{T}_\Theta$*

*Proof.* We begin by decomposing the error $\mathsf{err}_{P_{\mathsf{NL}}}(T_S)$ into three components as

$$\mathsf{err}_{P_{\mathsf{NL}}}(T_S) = \underbrace{\mathsf{err}_{P_{\mathsf{NL}}}(T_S) - \mathsf{err}_{P_{\mathsf{syn}}}(T_S)}_{\text{(I)}} + \underbrace{\mathsf{err}_{P_{\mathsf{syn}}}(T_S) - \hat{\mathsf{err}}_{P_{\mathsf{syn}}}(T_S)}_{\text{(II)}} + \hat{\mathsf{err}}_{P_{\mathsf{syn}}}(T_S). \tag{14}$$

We now upper bound each of the terms (I) and (II) separately.

**Bound on Term (I).** Let $\gamma$ denote an arbitrary joint distribution over the distributions $P_{\mathsf{syn}}$ and $P_{\mathsf{NL}}$. Then, we can bound the first term as

$$
\begin{aligned}
\mathrm{err}_{P_{\mathsf{NL}}}(T_S) - \mathrm{err}_{P_{\mathsf{syn}}}(T_S) &= \mathbb{E}_{P_{\mathsf{NL}}}[\ell(T(s), y)] - \mathbb{E}_{P_{\mathsf{syn}}}[\ell(T(s'), y')] \\
&\stackrel{(i)}{=} \mathbb{E}_\gamma[\ell(T(s), y) - \ell(T(s'), y')] \\
&\stackrel{(ii)}{\leq} \inf_\gamma \mathbb{E}_\gamma |\ell(T(s), y) - \ell(T(s'), y')| \ ,
\end{aligned}
\tag{15}
$$

where (i) follows from the independence of the two expectations and (ii) follows from that (i) holds for any arbitrary joint distribution $\gamma$. The final bound on this term now follows:

$$
\begin{aligned}
\inf_\gamma \mathbb{E}_\gamma |\ell(T(s), y) - \ell(T(s'), y')| &= \inf_\gamma \int |\ell(T(s), y) - \ell(T(s), y') + \ell(T(s), y') - \ell(T(s'), y')| \, d\gamma \\
&\stackrel{(i)}{\leq} c_1 \inf_\gamma \int |y - y'| - \|T(s') - T(s)\| d\gamma \\
&\stackrel{(ii)}{\leq} c_1 \max(1, L) \cdot \inf_\gamma \int |y - y'| - \|s' - s\| d\gamma \\
&= c_1 \max(1, L) \cdot W_1(P_{\mathsf{NL}}, P_{\mathsf{syn}}) \ ,
\end{aligned}
\tag{16}
$$

where the inequalities (i) follows from Assumption 1 and (ii) follows from Assumption 2. This completes the bound on Term (I).

**Bound on Term (II).** Using a standard generalization bound [34, see Theorem 26.5], we have with probability at least $1 - \delta$

$$
\begin{aligned}
\mathrm{err}_{P_{\mathsf{syn}}}(T_S) - \hat{\mathrm{err}}_{P_{\mathsf{syn}}}(T_S) &\leq \mathcal{R}(\ell \circ \mathcal{T}_\Theta) + 4\sqrt{\frac{2\ln(4/\delta)}{n_{\mathsf{syn}}}} \\
&\leq c_1 \cdot \mathcal{R}(\mathcal{T}_\Theta) + 4\sqrt{\frac{2\ln(4/\delta)}{n_{\mathsf{syn}}}} \\
&\stackrel{(i)}{\leq} c_1 \cdot \sqrt{\frac{2\mathrm{VC}(\mathcal{T}_\Theta)\ln m}{n_{\mathsf{syn}}}} + 4\sqrt{\frac{2\ln(4/\delta)}{n_{\mathsf{syn}}}}
\end{aligned}
\tag{17}
$$

where $\mathcal{R}(\ell \circ \mathcal{T}_\Theta)$ denotes the Rademacher complexity of the class $\mathcal{T}_\Theta$ composed with the loss function $\ell$ and inequality (i) follows from Sauer's Lemma.

Combining the bounds in equations (16) and (17) completes the proof of the theorem. $\square$

## D   Extension to multi-class classification

In this section, we present an extension of TART to the multi-class classification setup. Recall from Section 3.2 that the module was trained on synthetic binary tasks sampled from the logistic model as

$$
\text{Sequence } s_t : w_t \sim \mathcal{N}(0, I_d), \quad x_{i,t} \sim \mathcal{N}(0, I_d), \quad y_{i,t} \sim \sigma(\alpha\langle x_{i,t}, w_t\rangle) \qquad \text{for } i \in [k] \ .
\tag{18}
$$

Observe that each sequence corresponded to a single hyperplane $w_t$ which defined the binary classification problem. For the multi-class extension, suppose that there are $L$ labels. For each sequence $s_t$, we sample one hyperplane per class in $d$-dimensional space:

$$
w_{t,l} \sim \mathcal{N}(0, I_d) \quad \text{for } l \in [L] \ .
\tag{19}
$$

Each covariate $x_{i,t}$ is sampled again from the standard Normal distribution as before. The label $y_{i,t}$ is sampled from the multinomial distribution over $[L]$ given by:

$$
\mathbb{P}(y_{i,t} = l) \propto \exp(\langle x_{i,t}, w_{t,l}\rangle).
\tag{20}
$$

Given this distribution over sequences $s_t$, we train the TART module as before, but with the linear layer with output dimension $L$ instead of 1 as with the binary classification setup. We evaluate our

| Dataset | Num Labels | Test Size | Max char length | Max token length | Avg. char length | Avg. token length |
|---|---|---|---|---|---|---|
| AGNews | 4 | 2000 | 814 | 238 | 236.97 | 52.31 |
| Ecommerce | 4 | 2000 | 50328 | 10544 | 678.768 | 158.02 |
| DBPedia-14 | 14 | 1988 | 691 | 240 | 284.08 | 67.52 |

Table 1: Dataset (test) statistics for multi-class datasets.

| Dataset | Method | $k = 18$ | $k = 32$ | $k = 48$ | $k = 64$ |
|---|---|---|---|---|---|
| AGNews | TART | 63.55 (+38.5) | 64.45 (+39.6) | 71.15 (+45.9) | 72.35 (+45.4) |
| | ICL | 25.1 | 24.9 | 25.3 | 27.0 |
| Ecommerce | TART | 78.2 (+52.3) | 79.4 (+52.6) | 75.4 (+47.1) | 79.3 (+50.8) |
| | ICL | 25.1 | 24.9 | 25.3 | 27.0 |
| DBPedia-14 | TART | 52.4 (+45.1) | 63.8 (+52.8) | 63.1 (+49.5) | 68.4 (+59.3) |
| | ICL | 7.3 | 11.0 | 13.6 | 9.1 |

Table 2: **TART vs. Base ICL Performance on multi-class classification tasks**. Evaluations of GPT-Neo-125M model in the multi-class setup. $k$ = Number of in-context examples.

multi-class variant of TART on AGNews-4, Ecommerce-4 and the DBPedia-14 datasets (see Table 1 for dataset statistics).

Table 2 shows the evaluation our proposed multi-class extension of TART on the above three datasets. While the in-context learning performance of the base model (GPT-Neo-125M) is close to random, we see that the TART substantially improves upon this performance. It provides improvement of up to 45.9% on AGNews, 52.6% on Ecommerce, and 59.3% on DBPedia-14 datasets.

# E   Details for experimental evaluation

We describe supplementary experimental details from Section 4 as well as additional results for the natural language benchmark evaluations (Section E.2) and results for other modalities (vision and audio) (Section E.5).

## E.1   Experimental setup

We begin by providing dataset statistics and details of the baselines.

### E.1.1   Dataset construction and statistics

Table 4 and 3 provides a detailed breakdown of dataset statistics. For each dataset, we use the original test sets with the exception of Civil Comments [8], AG News [48] and DBPedia [48]. For the multi-class datasets—AG News [48] and DBPedia [48] — we construct 4 binary classification tasks for each datasets. More concretely, AG News labels news articles into four categories: World, Sports, Business, and Science/Technology. We create a separate binary classification task for each category, sampling negatives from the remaining classes. DBPedia is a 14-way ontology classification dataset. We create 4 separate binary classification tasks for the educational institution, company, artist, and athlete ontologies, sampling negatives from the remaining classes. For the train set, we sample a class-balanced set of 64 examples from the original dataset. For each dataset, we sample 5 separate training sets, using 5 different random seeds. In evaluations, we evaluate TART and the baseline methods across each of these 5 different training sets.

| Dataset | Test size | Max char length | Max token length | Avg. char length | Avg. token length |
|---|---|---|---|---|---|
| AG-News-0 | 3800 | 732 | 259 | 237.07 | 51.36 |
| AG-News-1 | 3800 | 814 | 213 | 232.01 | 51.46 |
| AG-News-2 | 3800 | 814 | 225 | 236.10 | 52.25 |
| AG-News-3 | 3800 | 892 | 259 | 234.86 | 51.38 |
| Civil Comments | 11576 | 1000 | 634 | 272.72 | 61.73 |
| DBPedia-0 | 10000 | 2081 | 629 | 300.94 | 65.83 |
| DBPedia-1 | 10000 | 2081 | 629 | 298.81 | 66.91 |
| DBPedia-2 | 10000 | 2081 | 883 | 286.85 | 66.53 |
| DBPedia-3 | 10000 | 2081 | 629 | 275.81 | 63.88 |
| IMDB | 25000 | 12988 | 2972 | 1293.79 | 292.82 |
| Rotten Tomatoes | 1066 | 261 | 63 | 115.52 | 25.36 |
| SMS Spam | 4181 | 612 | 258 | 81.46 | 23.76 |
| SST | 2210 | 256 | 60 | 102.40 | 22.34 |
| Youtube | 250 | 1125 | 292 | 112.50 | 31.84 |

Table 3: Dataset (test) statistics for all NLP datasets.

| Dataset | Max char length | Max token length | Avg. char length | Avg. token length |
|---|---|---|---|---|
| AG-News-0 | 701 | 256 | 236.15 | 51.53 |
| AG-News-1 | 749 | 180 | 232.48 | 51.61 |
| AG-News-2 | 735 | 256 | 241.70 | 53.91 |
| AG-News-3 | 1002 | 258 | 241.21 | 53.21 |
| Civil Comments | 1000 | 347 | 280.97 | 63.44 |
| DBPedia-0 | 707 | 207 | 300.48 | 65.79 |
| DBPedia-1 | 1023 | 280 | 299.89 | 66.57 |
| DBPedia-2 | 628 | 203 | 288.21 | 66.22 |
| DBPedia-3 | 758 | 203 | 279.45 | 64.24 |
| IMDB | 7068 | 1630 | 1284.20 | 290.00 |
| Rotten Tomatoes | 260 | 62 | 112.46 | 24.82 |
| SMS Spam | 911 | 217 | 106.90 | 31.87 |
| SST | 248 | 56 | 101.97 | 22.34 |
| Youtube | 1089 | 767 | 90.12 | 29.98 |

Table 4: Dataset (train) statistics for all NLP datasets.

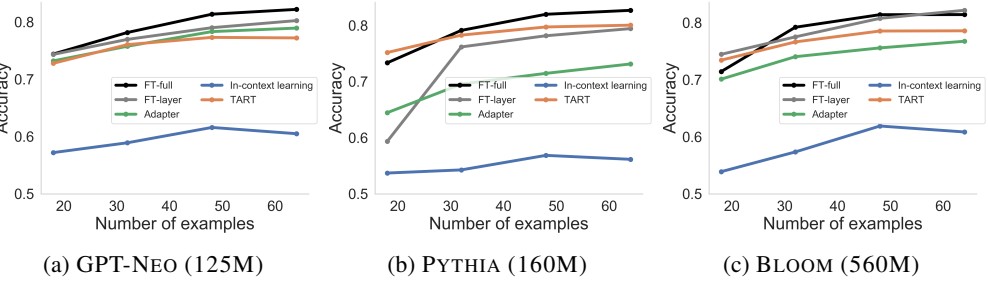

(a) GPT-NEO (125M)  (b) PYTHIA (160M)  (c) BLOOM (560M)

Figure 19: **Comparison of all methods**. TART significantly improves base in-context learning performance and is competitive with full-finetuning across model families.

### E.1.2 Baseline methods

For each dataset, we compare TART to 4 baseline task-adaptation methods: 1) in-context learning, 2) full fine-tuning, 3) last layer fine-tuning, and 4) adapters. The last layer fine-tuning and the adapters are trained as follows:

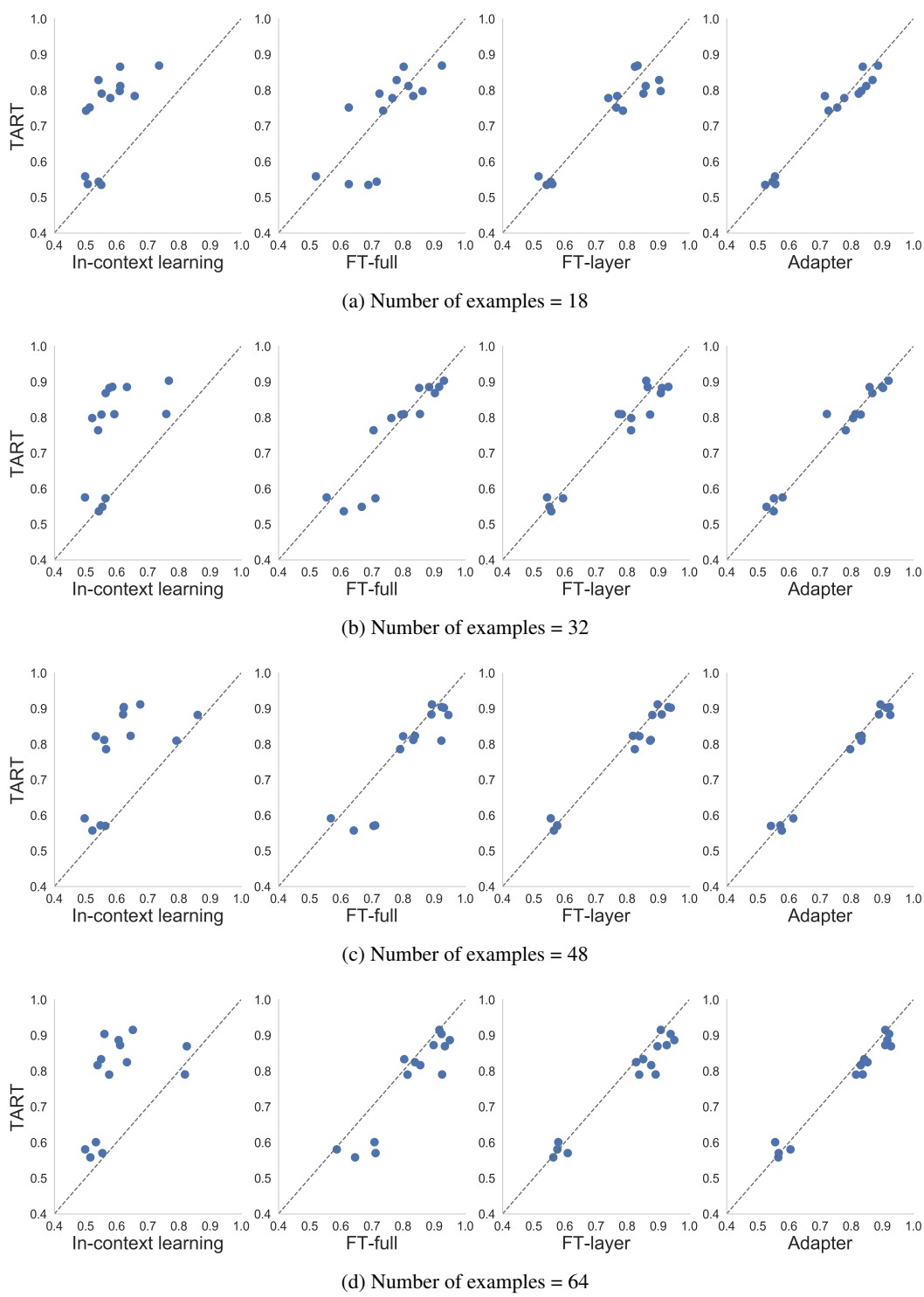

Figure 20: **Comparison of TART and task-adaptation approaches (GPT-NEO (125M))**. We see that for GPT-NEO (125M), TART outperforms in-context learning and is competitive with full fine-tuning and adapters across all $k$.

- Last layer fine-tuning: Freeze all layers of transformer but the final transformer block and the language modeling head.

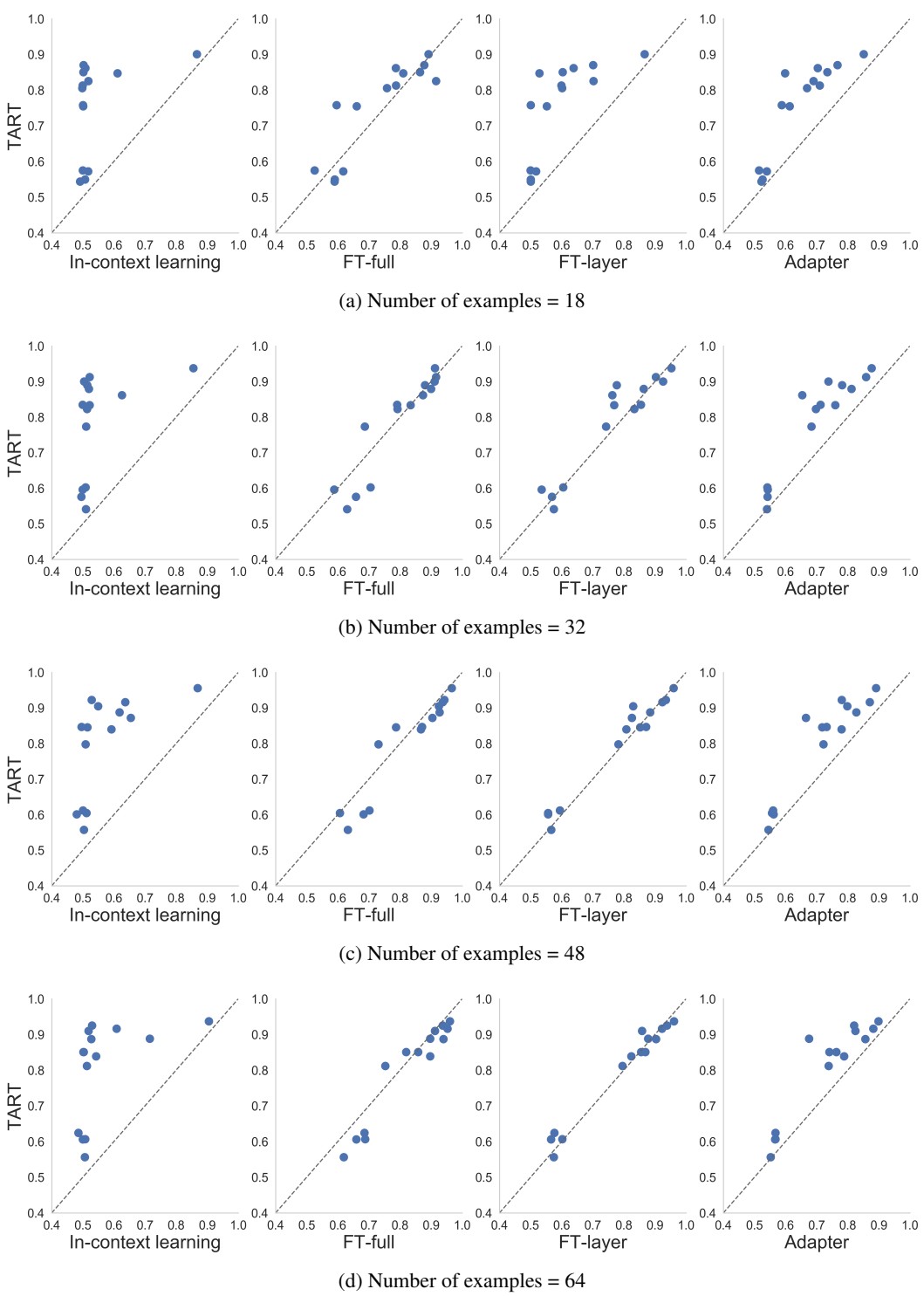

Figure 21: **Comparison of TART and task-adaptation approaches (PYTHIA (160M))**. We see that for PYTHIA (160M), TART outperforms in-context learning and adapters and is competitive with full fine-tuning across all $k$.

- Adapter: Combine a frozen LLM base transformer model with a trainable adapter head—an MLP composed of a single linear layer followed by non-linearity.

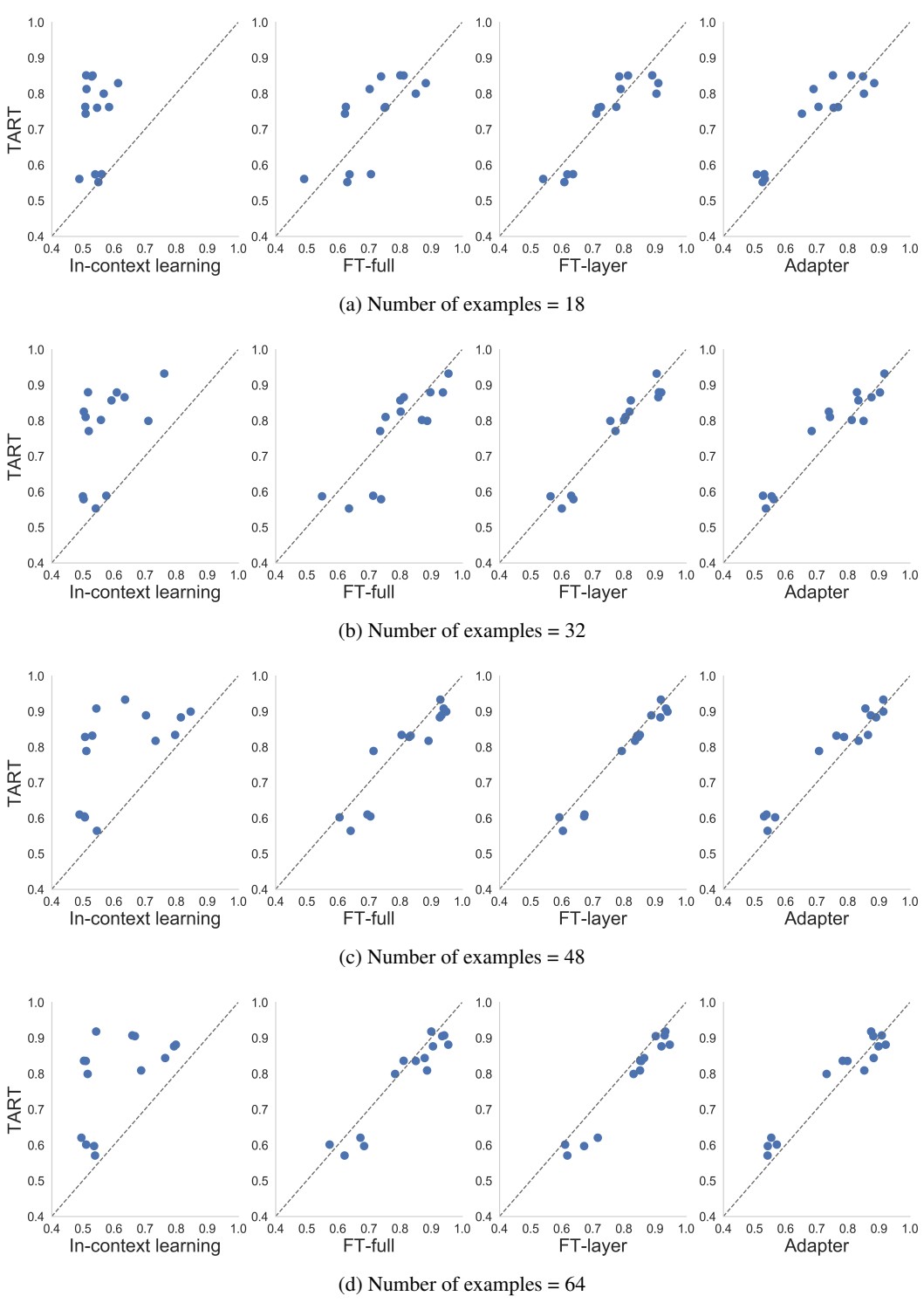

Figure 22: **Comparison of TART and task-adaptation approaches (BLOOM (560M))**. We see that for BLOOM (560M), TART outperforms in-context learning and adapters and is competitive with full fine-tuning across all $k$.

**Hyperparameter search.** For each baseline, we perform an extensive hyperparameter search over number of epochs and learning rate for each dataset in order to optimize performance. We search over a range of learning rates (1e-3, 1e-4, 3e-5, 1e-5, 8e-6), and range of epochs (5, 10, 15, 20, 50). For all models < 1B parameters, we use a batch size of 1. For all models > 1B parameters, we use a batch

| Model | TART | GPT-J (6B) | OPT (175B) | BLOOM (176B) | GPT-3 (175B) |
|---|---|---|---|---|---|
| Accuracy | 0.634 | 0.608 | 0.637 | 0.595 | 0.673 |

Table 5: **RAFT (HELM) Binary Classification Performance (Average Accuracy)**. TART is used with GPT-NEO (125M) model which is 1000x smaller than the corresponding 175B parameter models. TART outperforms BLOOM (176B) and is competitive with OPT (175B) and GPT-3 (175B).

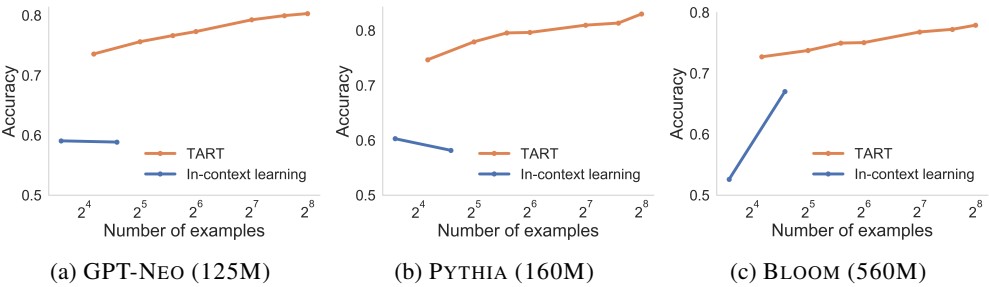

(a) GPT-NEO (125M)    (b) PYTHIA (160M)    (c) BLOOM (560M)

Figure 23: **Beyond context window constraints**. Performance comparison with respect to number of in-context examples. Base in-context learning is bound with respect to total numbers of examples and performance saturates. TART is not bound by context length, and performance continues to scale as number of examples increases.

size of 8. We use these same batch sizes at evaluation time. We perform our hyperparameter searches with a fixed number of train samples (64). We run our hyperparameter searches over 3 random seeds.

### E.2 NL benchmarks

In this section, we provide additional results deferred from Section 4 on the NLP benchmark evaluations, RAFT evaluations and demonstration of TART's data-scalability.

#### E.2.1 Performance on benchmark datasets

Figure 19 shows the performance of the baseline methods with TART averaged across the suite of 14 datasets. TART, while being task-agnostic, shows similar performance quality to task-specific approaches across the different model families, and consistently outperforms in-context learning.

Figures 20, 21, and 22 show the scatter plots of the accuracies of TART with the baseline methods across datasets and different values of in-context examples $k$. An interesting observation is that as the number of examples $k$ increases from 18 to 64, the performance of fine-tuning improves at a better rate than that of TART.

#### E.2.2 Real-world Annotated Few-shot Tasks (RAFT) evaluation

For our evaluations on the RAFT benchmark [2], we follow the protocol (same train and test sets) used in HELM benchmark. The HELM benchmark [22] contains the evaluation results for many open and closed models enabling us to accurately compare the performance of TART with other models. We evaluate TART on all RAFT binary classification datasets (twitter-complaints, neurips-impact-statement-risks, overulling, ade-corpusv2, tweet-eval-hate, terms-of-service, tai-safety-research) with the exception of systematic-review-inclusion which contains zero positive samples in the train set. TART requires at least one example of each class in the training set. Table 5 contains a detailed performance comparison of TART with respect to other models. TART when combined with GPT-NEO (125M) is able to outperform BLOOM (176B) and is competitive with OPT (175B) and GPT-3 (175B), all of which have 1000x more parameters.

#### E.2.3 Beyond context length: TART is data-scalable

**Setup.** For these evaluations, we use the a subset of 6 datasets: AG-News-0, DBPedia-0, SST, SMS Spam, Youtube and Rotten Tomatoes. We evaluate the performance of TART over $k$=[18, 32, 48, 64, 128, 192, 256] where $k$ is the number of in-context examples. When evaluating our base models, we

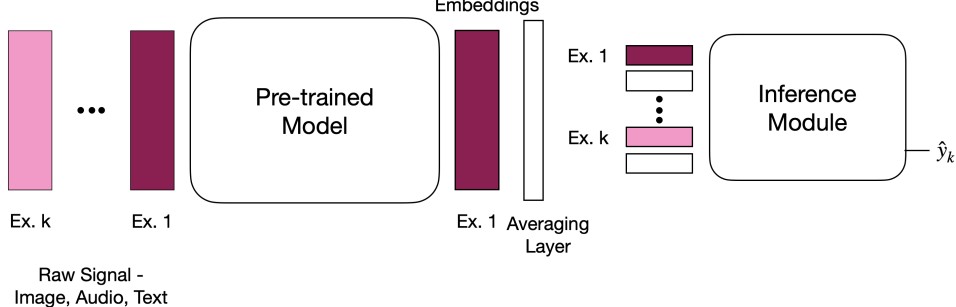

Figure 24: **Stream embeddings**. Another protocol for generating representations for in-context examples where each example is embedded by the base model separately.

evaluate over $k$=[8, 24]—values of $k$ that maximize the context window. We use a lower-bound of 8 given that the maximum input sequence length in the training set for AG News is 256. With such a sequence length, the maximum number of in-context examples that fit in the context-window is 8, hence the lower bound.

**Embeddings.**    For these evaluations, we use what we call "streaming" embeddings (see Figure 24). In this setup, we use the context window of the LLM to encode a single example at a time. The final embeddings are then averaged and used in-context with TART's reasoning module. This is in contrast to the vanilla and LOO embeddings which use multiple examples in-context with the base LLM to obtain the embeddings.

**Evaluation.**    Figure 23 shows the performance of base in-context learning with TART across the three different model families. Observe that while in-context learning is bottlenecked by the context window of the base LLM, TART is able to learn from 10x more examples and exhibits an increasing trend in accuracy with number of examples across models.

### E.3    Generalization across architectures

In this section, we demonstrate that it is possible to train TART reasoning modules on architectures beyond transformers. More concretely, we train a reasoning module that has the Hyena architecture [30] using the same synthetic logistic regression tasks.

### E.3.1    Setup and Hyper-parameters

We instantiate a reasoning module with 12 Hyena blocks, a hidden dimension size of 256, and a sequence length of 2050. We train with a batch size of 16, using a learning rate of 5e-05. We sample data with a noise parameter ($\alpha$) equal to 1. We train the model for 5000 steps. For our evaluations, we use the final checkpoint (i.e., 5000) of the reasoning module.

### E.3.2    Evaluation

We evaluate the Hyena-based reasoning module applied to GPT-NEO (125M) on 6 datasets: SMS Spam, SST, AG-News-0, DBpedia-14-0, Youtube, and Rotten Tomatoes. As seen in Figure 25, the Hyena-based reasoning module transfers to natural language tasks and performs competitively with the transformer-based reasoning module

### E.4    Performance in low sample regime

In Figure 26, we evaluate our models in the low-sample regime where the number of in-context examples range from $k = 4 - 10$. TART outperforms both in-context learning and task-specific fine-tuning baselines.

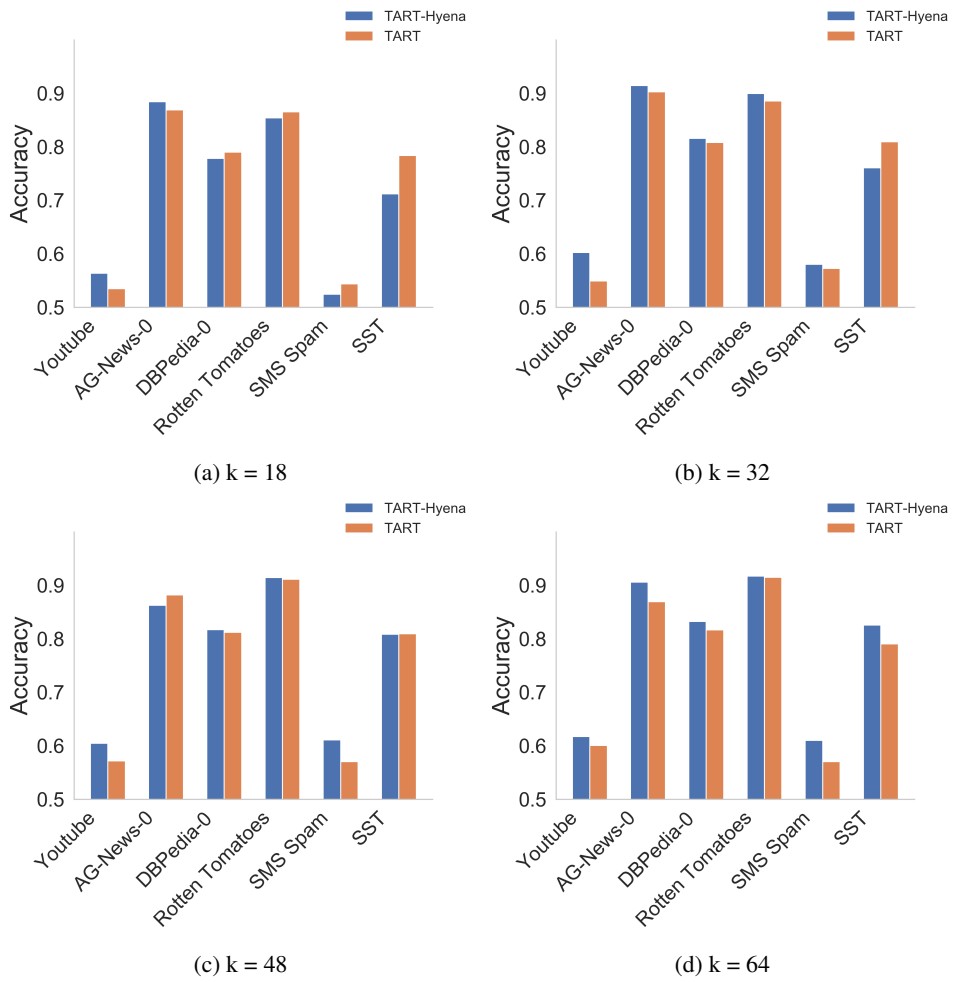

(a) k = 18

(b) k = 32

(c) k = 48

(d) k = 64

Figure 25: **Comparing TART with a Hyena version of TART.** Instead of using a Transformer module, we use a Hyena model to learn the probabilistic reasoning task. Hyena-TART is comparable to TART across datasets and achieves better performance for larger value of $k$ ($= 64$).

## E.5 Extension to other modalities: TART is domain-agnostic!

We begin by providng a description of the datasets we used to evaluate TART on audio and vision tasks, and then provide additional results comparing our algorithm with baselines.

### E.5.1 Dataset details

For audio classification, we use the Speech Commands (Version 0.01) dataset [40]. Speech Commands is a multi-class classification task where the task is to detect preregistered keywords by classifying utterances into a predefined set of words. We construct a 3 binary classification task over the keywords "stop" and "go", "up" and "down", and "yes" and "no" (see Table 6 for more details).

For image classification, we use CIFAR-10 [17] and MNIST [19]. Both tasks are multi-class classification tasks. We create 3 binary classification tasks for each of the datasets. For CIFAR-10 the tasks are: airplane vs. bird, bird vs. horse, and ship vs. automobile. For MNIST the tasks are: 0 vs. 8, 1 vs. 6 and 2 vs. 4. See Table 6 for more details.

For both the audio and image datasets, we sample a class-balanced set of 256 samples from the training set. For the test sets, we filter the original test sets to only include samples of the two classes we are learning to predict for (i.e., airplane and bird for CIFAR10 and 0 and 8 for MNIST).

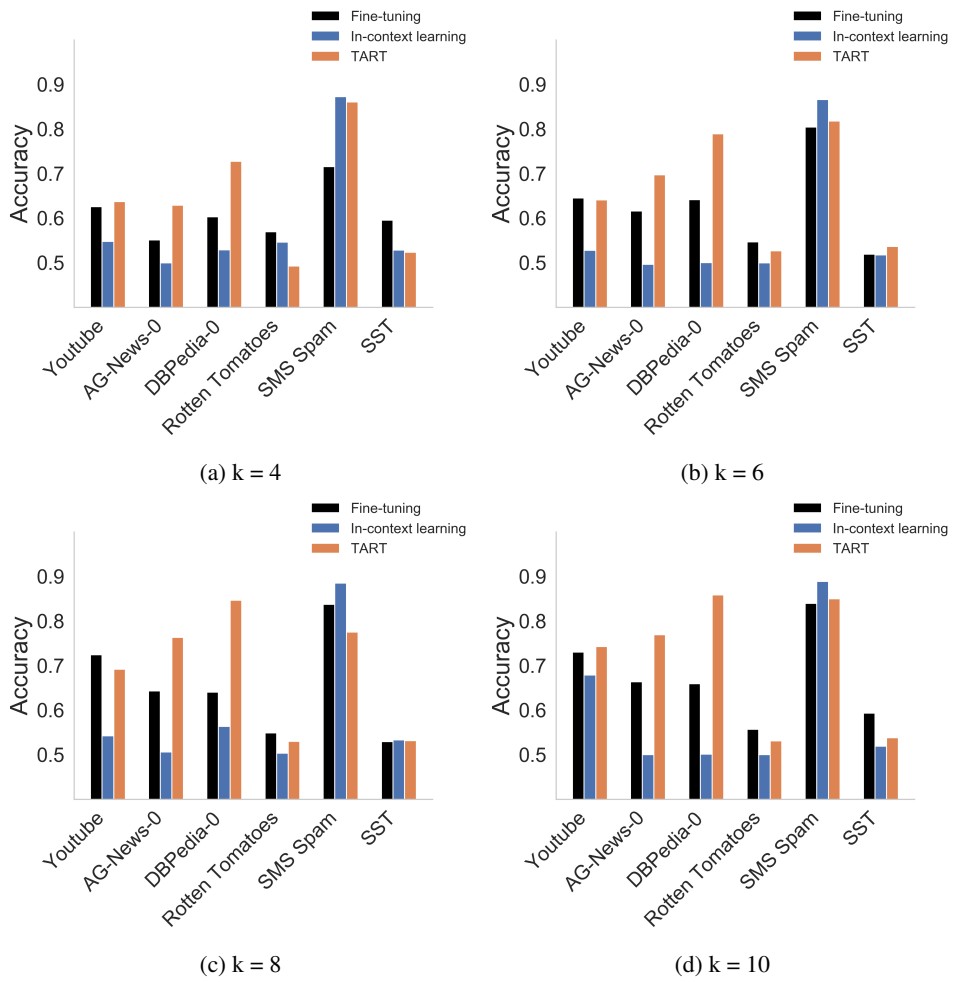

Figure 26: **TART in low sample regime**

| Dataset | Modality | Train size | Test size |
|---|---|---|---|
| MNIST (0 vs. 8) | image | 256 | 1954 |
| MNIST (1 vs. 6) | image | 256 | 1940 |
| MNIST (2 vs. 4) | image | 256 | 2014 |
| Speech Commands (stop vs. go) | audio | 256 | 500 |
| Speech Commands (up vs. down) | audio | 256 | 508 |
| Speech Commands (yes vs. no) | audio | 256 | 525 |
| CIFAR-10 (airplane vs. bird) | image | 256 | 2000 |
| CIFAR-10 (bird vs. horse) | image | 256 | 2000 |
| CIFAR-10 (ship vs. automobile) | image | 256 | 2000 |

Table 6: Dataset statistics for all audio and image evaluation datasets.

### E.5.2    Algorithms for comparison

For these evaluations, we use the "streaming embeddings" described in Figure 24 to obtain the embedding for TART. We evaluate over $k$=[18, 32, 48, 64, 128, 256].

We compare against two baseline task-adaptation methods: 1) full fine-tuning and 2) adapters. We use the same architectures as described in Appendix E.1.2. For vision tasks, we use Google's 307M

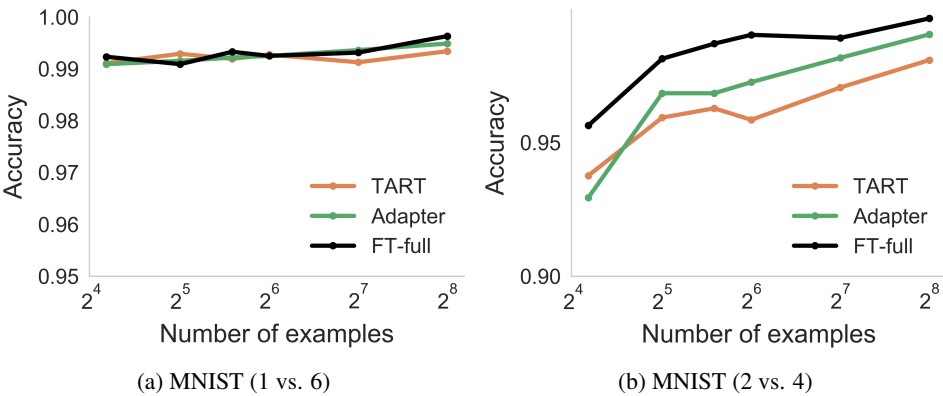

(a) MNIST (1 vs. 6)  (b) MNIST (2 vs. 4)

Figure 27: **Additional MNIST binary classification tasks**. TART is competitive with task-specific full fine-tuning and adapters.

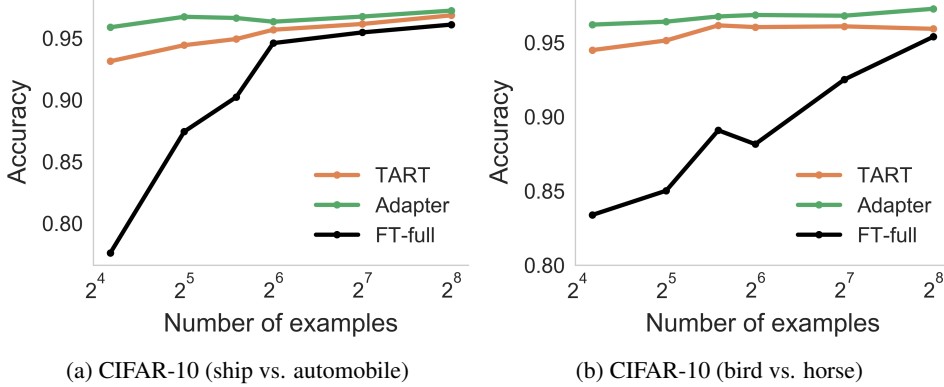

(a) CIFAR-10 (ship vs. automobile)  (b) CIFAR-10 (bird vs. horse)

Figure 28: **Additional CIFAR-10 binary classification tasks**. TART is competitive with task-specific full fine-tuning and adapters.

parameter pretrained Vision Transformer (ViT) model [44]: VIT-LARGE-PATCH16-224. For audio tasks, we use OpenAI's 1.5B parameter pretrained Whisper model [31]: WHISPER-LARGE.

**Hyperparameter search**   For each baseline, we perform an extensive hyperparameter search over number of epochs and learning rate for each dataset in order to optimize performance. We search over a range of learning rates (1e-3, 5e-04, 1e-4, 5e-5, 1e-5, and 8e-6) and a range of epochs (5, 10, 15 and 20). For all models we use a batch size of 1. We perform our hyperparameter searches for a fixed number of train samples (128) and run our hyperparameter searches over 3 random seeds.

### E.5.3   Evaluation

We plot the accuracy as a function of the number of examples for TART, fine-tuning and adapter in Figure 27 (MNIST), Figure 28 (CIFAR-10), and Figure 29 (Speech Commands). TART is competitive with both these baselines, showing how task-agnostic methods can compete with task-specific adaptation methods across different modalities.

## F   Broader Impact

Our submission focuses on understanding in-context learning and how it compares to task-specific fine-tuning approaches. We view our work as a *general understanding* paper and, to the best of our knowledge, see no negative consequences of our work. We hope that our work furthers the state of understanding of large language models (LLMs). Furthermore, because TART is a plug-and-play solution that doesn't require *any* fine-tuning, we hope that our method can help domain experts (e.g., lawyers, scientists) incorporate LLMs into their workflows. In this sense, we view TART as a

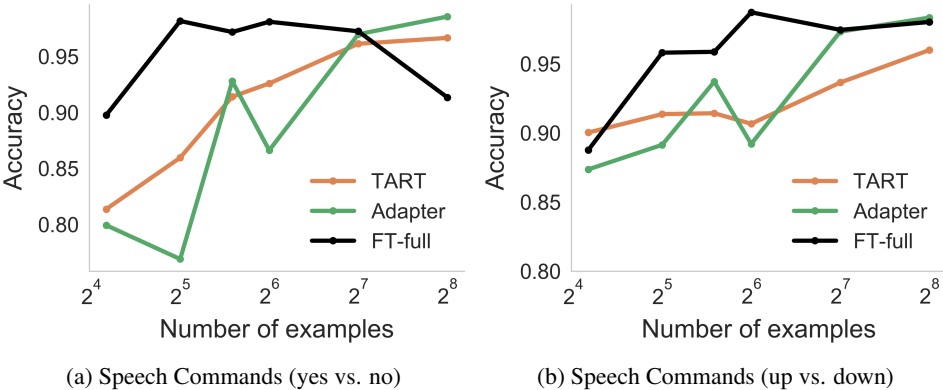

(a) Speech Commands (yes vs. no)    (b) Speech Commands (up vs. down)

Figure 29: **Additional Speech Commands binary classification tasks**. TART is competitive with task-specific full fine-tuning and adapters.

contribution that is democratizing ML by making them more accessible and usable to experts in a variety of domains.

## G  Compute Resource Estimates

**Resources for training TART**  We use a single NVIDIA RTX A6000 GPU ( $2/hr) for  6 hours to train our TART inference head, costing a total of $18 to train.

**Resources for training task-specific baselines**  We use 4 NVIDIA A100 GPU's ( $3.5 per GPU/hr) for  100 hours for hyperparamter tuning, costing a total of $1,400. We use 8 NVIDIA A100 GPU's ( $3.5 per GPU/hr) for  50 hours to fine-tune all task-adaptation baseline models. Total cost for fine-tuning amounted to $1,440.

**Resources for inference**  We use 8 NVIDIA A100 GPU's ( $3.5 per GPU/hr) for  480 hours for all inference runs costing a total of $13,440.

| Dataset | TART | In-context learning | Fine-tuning full | Fine-tuning layer | Adapters |
|---|---|---|---|---|---|
| AG-News-0 | $0.790 \pm 0.036$ | $0.552 \pm 0.049$ | $0.724 \pm 0.070$ | $0.852 \pm 0.019$ | $0.824 \pm 0.031$ |
| AG-News-1 | $0.828 \pm 0.021$ | $0.541 \pm 0.037$ | $0.779 \pm 0.137$ | $0.903 \pm 0.021$ | $0.868 \pm 0.020$ |
| AG-News-2 | $0.751 \pm 0.023$ | $0.513 \pm 0.014$ | $0.626 \pm 0.057$ | $0.765 \pm 0.025$ | $0.755 \pm 0.012$ |
| AG-News-3 | $0.743 \pm 0.031$ | $0.502 \pm 0.017$ | $0.736 \pm 0.035$ | $0.786 \pm 0.025$ | $0.727 \pm 0.066$ |
| Civil Comments | $0.559 \pm 0.027$ | $0.499 \pm 0.002$ | $0.520 \pm 0.033$ | $0.515 \pm 0.019$ | $0.555 \pm 0.036$ |
| DBPedia-0 | $0.866 \pm 0.030$ | $0.611 \pm 0.091$ | $0.802 \pm 0.020$ | $0.825 \pm 0.012$ | $0.837 \pm 0.022$ |
| DBPedia-1 | $0.778 \pm 0.036$ | $0.579 \pm 0.100$ | $0.766 \pm 0.056$ | $0.740 \pm 0.041$ | $0.778 \pm 0.044$ |
| DBPedia-2 | $0.798 \pm 0.042$ | $0.609 \pm 0.136$ | $0.862 \pm 0.041$ | $0.908 \pm 0.011$ | $0.832 \pm 0.048$ |
| DBPedia-3 | $0.812 \pm 0.032$ | $0.611 \pm 0.135$ | $0.817 \pm 0.034$ | $0.859 \pm 0.025$ | $0.848 \pm 0.028$ |
| IMDB | $0.537 \pm 0.022$ | $0.507 \pm 0.007$ | $0.625 \pm 0.013$ | $0.560 \pm 0.021$ | $0.556 \pm 0.014$ |
| Rotten Tomatoes | $0.535 \pm 0.030$ | $0.550 \pm 0.043$ | $0.689 \pm 0.019$ | $0.541 \pm 0.037$ | $0.524 \pm 0.018$ |
| SMS Spam | $0.869 \pm 0.063$ | $0.736 \pm 0.099$ | $0.925 \pm 0.011$ | $0.833 \pm 0.015$ | $0.886 \pm 0.023$ |
| SST | $0.544 \pm 0.021$ | $0.542 \pm 0.024$ | $0.715 \pm 0.009$ | $0.555 \pm 0.026$ | $0.547 \pm 0.009$ |
| Youtube | $0.784 \pm 0.047$ | $0.658 \pm 0.089$ | $0.833 \pm 0.089$ | $0.768 \pm 0.100$ | $0.715 \pm 0.136$ |

Table 7: **Standard deviation of accuracy, number of examples = 18, GPT-NEO (125M)**

| Dataset | TART | In-context learning | Fine-tuning full | Fine-tuning layer | Adapters |
|---|---|---|---|---|---|
| AG-News-0 | $0.805 \pm 0.029$ | $0.498 \pm 0.002$ | $0.758 \pm 0.127$ | $0.601 \pm 0.072$ | $0.669 \pm 0.044$ |
| AG-News-1 | $0.825 \pm 0.023$ | $0.517 \pm 0.019$ | $0.916 \pm 0.015$ | $0.702 \pm 0.097$ | $0.690 \pm 0.034$ |
| AG-News-2 | $0.758 \pm 0.027$ | $0.500 \pm 0.000$ | $0.596 \pm 0.031$ | $0.500 \pm 0.001$ | $0.588 \pm 0.026$ |
| AG-News-3 | $0.754 \pm 0.033$ | $0.501 \pm 0.003$ | $0.661 \pm 0.093$ | $0.552 \pm 0.041$ | $0.613 \pm 0.022$ |
| Civil Comments | $0.575 \pm 0.019$ | $0.500 \pm 0.003$ | $0.525 \pm 0.049$ | $0.500 \pm 0.000$ | $0.515 \pm 0.008$ |
| DBPedia-0 | $0.861 \pm 0.018$ | $0.508 \pm 0.016$ | $0.786 \pm 0.054$ | $0.638 \pm 0.109$ | $0.704 \pm 0.038$ |
| DBPedia-1 | $0.813 \pm 0.020$ | $0.499 \pm 0.010$ | $0.787 \pm 0.067$ | $0.599 \pm 0.078$ | $0.710 \pm 0.059$ |
| DBPedia-2 | $0.870 \pm 0.035$ | $0.502 \pm 0.025$ | $0.878 \pm 0.071$ | $0.701 \pm 0.113$ | $0.767 \pm 0.052$ |
| DBPedia-3 | $0.850 \pm 0.050$ | $0.502 \pm 0.003$ | $0.864 \pm 0.034$ | $0.603 \pm 0.114$ | $0.734 \pm 0.030$ |
| IMDB | $0.550 \pm 0.027$ | $0.507 \pm 0.005$ | $0.590 \pm 0.046$ | $0.500 \pm 0.000$ | $0.526 \pm 0.012$ |
| Rotten Tomatoes | $0.544 \pm 0.027$ | $0.491 \pm 0.013$ | $0.589 \pm 0.074$ | $0.500 \pm 0.000$ | $0.522 \pm 0.023$ |
| SMS Spam | $0.901 \pm 0.038$ | $0.867 \pm 0.030$ | $0.892 \pm 0.052$ | $0.867 \pm 0.004$ | $0.851 \pm 0.042$ |
| SST | $0.572 \pm 0.016$ | $0.517 \pm 0.002$ | $0.617 \pm 0.074$ | $0.517 \pm 0.000$ | $0.539 \pm 0.007$ |
| Youtube | $0.847 \pm 0.047$ | $0.611 \pm 0.084$ | $0.810 \pm 0.060$ | $0.528 \pm 0.000$ | $0.598 \pm 0.103$ |

Table 8: **Standard deviation of accuracy, number of examples = 18, PYTHIA (160M)**

| Dataset | TART | In-context learning | Fine-tuning full | Fine-tuning layer | Adapters |
|---|---|---|---|---|---|
| AG-News-0 | $0.813 \pm 0.021$ | $0.511 \pm 0.013$ | $0.702 \pm 0.102$ | $0.790 \pm 0.033$ | $0.689 \pm 0.060$ |
| AG-News-1 | $0.851 \pm 0.020$ | $0.511 \pm 0.014$ | $0.801 \pm 0.086$ | $0.891 \pm 0.037$ | $0.752 \pm 0.032$ |
| AG-News-2 | $0.744 \pm 0.034$ | $0.509 \pm 0.009$ | $0.622 \pm 0.083$ | $0.711 \pm 0.070$ | $0.652 \pm 0.047$ |
| AG-News-3 | $0.763 \pm 0.026$ | $0.508 \pm 0.014$ | $0.626 \pm 0.016$ | $0.775 \pm 0.035$ | $0.706 \pm 0.027$ |
| Civil Comments | $0.561 \pm 0.029$ | $0.489 \pm 0.009$ | $0.491 \pm 0.032$ | $0.540 \pm 0.029$ | $0.533 \pm 0.030$ |
| DBPedia-0 | $0.851 \pm 0.009$ | $0.531 \pm 0.044$ | $0.811 \pm 0.116$ | $0.813 \pm 0.057$ | $0.812 \pm 0.043$ |
| DBPedia-1 | $0.760 \pm 0.037$ | $0.546 \pm 0.088$ | $0.750 \pm 0.129$ | $0.718 \pm 0.067$ | $0.754 \pm 0.041$ |
| DBPedia-2 | $0.800 \pm 0.032$ | $0.567 \pm 0.110$ | $0.850 \pm 0.086$ | $0.904 \pm 0.018$ | $0.851 \pm 0.055$ |
| DBPedia-3 | $0.848 \pm 0.025$ | $0.528 \pm 0.046$ | $0.739 \pm 0.133$ | $0.785 \pm 0.132$ | $0.849 \pm 0.011$ |
| IMDB | $0.552 \pm 0.031$ | $0.550 \pm 0.044$ | $0.630 \pm 0.016$ | $0.608 \pm 0.014$ | $0.526 \pm 0.025$ |
| Rotten Tomatoes | $0.574 \pm 0.029$ | $0.539 \pm 0.031$ | $0.638 \pm 0.037$ | $0.618 \pm 0.049$ | $0.507 \pm 0.010$ |
| SMS Spam | $0.830 \pm 0.112$ | $0.613 \pm 0.261$ | $0.883 \pm 0.130$ | $0.911 \pm 0.035$ | $0.885 \pm 0.045$ |
| SST | $0.574 \pm 0.019$ | $0.561 \pm 0.062$ | $0.707 \pm 0.024$ | $0.636 \pm 0.025$ | $0.531 \pm 0.015$ |
| Youtube | $0.762 \pm 0.116$ | $0.584 \pm 0.144$ | $0.753 \pm 0.140$ | $0.726 \pm 0.052$ | $0.769 \pm 0.084$ |

Table 9: **Standard deviation of accuracy, number of examples = 18, BLOOM (560M)**

| Dataset | TART | In-context learning | Fine-tuning full | Fine-tuning layer | Adapters |
|---|---|---|---|---|---|
| AG-News-0 | 0.808 ± 0.030 | 0.551 ± 0.041 | 0.795 ± 0.049 | 0.874 ± 0.022 | 0.830 ± 0.034 |
| AG-News-1 | 0.883 ± 0.014 | 0.577 ± 0.055 | 0.852 ± 0.070 | 0.911 ± 0.021 | 0.902 ± 0.011 |
| AG-News-2 | 0.764 ± 0.019 | 0.540 ± 0.027 | 0.705 ± 0.063 | 0.812 ± 0.019 | 0.782 ± 0.019 |
| AG-News-3 | 0.798 ± 0.025 | 0.521 ± 0.039 | 0.762 ± 0.046 | 0.813 ± 0.012 | 0.806 ± 0.028 |
| Civil Comments | 0.575 ± 0.054 | 0.498 ± 0.002 | 0.554 ± 0.029 | 0.543 ± 0.022 | 0.579 ± 0.044 |
| DBPedia-0 | 0.886 ± 0.021 | 0.632 ± 0.096 | 0.884 ± 0.019 | 0.866 ± 0.013 | 0.859 ± 0.020 |
| DBPedia-1 | 0.809 ± 0.031 | 0.593 ± 0.062 | 0.802 ± 0.030 | 0.783 ± 0.022 | 0.813 ± 0.023 |
| DBPedia-2 | 0.886 ± 0.011 | 0.586 ± 0.078 | 0.916 ± 0.029 | 0.932 ± 0.013 | 0.899 ± 0.017 |
| DBPedia-3 | 0.868 ± 0.022 | 0.565 ± 0.041 | 0.902 ± 0.016 | 0.908 ± 0.012 | 0.868 ± 0.023 |
| IMDB | 0.537 ± 0.040 | 0.543 ± 0.029 | 0.609 ± 0.029 | 0.556 ± 0.024 | 0.551 ± 0.036 |
| Rotten Tomatoes | 0.549 ± 0.035 | 0.554 ± 0.036 | 0.667 ± 0.036 | 0.550 ± 0.045 | 0.528 ± 0.020 |
| SMS Spam | 0.903 ± 0.027 | 0.768 ± 0.105 | 0.931 ± 0.014 | 0.861 ± 0.027 | 0.920 ± 0.011 |
| SST | 0.573 ± 0.009 | 0.564 ± 0.045 | 0.711 ± 0.022 | 0.594 ± 0.022 | 0.551 ± 0.022 |
| Youtube | 0.810 ± 0.072 | 0.759 ± 0.074 | 0.854 ± 0.044 | 0.773 ± 0.052 | 0.722 ± 0.113 |

Table 10: **Standard deviation of accuracy, number of examples = 32, GPT-NEO (125M)**

| Dataset | TART | In-context learning | Fine-tuning full | Fine-tuning layer | Adapters |
|---|---|---|---|---|---|
| AG-News-0 | 0.835 ± 0.020 | 0.499 ± 0.002 | 0.791 ± 0.100 | 0.855 ± 0.017 | 0.712 ± 0.028 |
| AG-News-1 | 0.900 ± 0.012 | 0.504 ± 0.004 | 0.911 ± 0.022 | 0.926 ± 0.009 | 0.738 ± 0.028 |
| AG-News-2 | 0.773 ± 0.012 | 0.510 ± 0.006 | 0.687 ± 0.034 | 0.742 ± 0.044 | 0.683 ± 0.025 |
| AG-News-3 | 0.823 ± 0.024 | 0.514 ± 0.017 | 0.792 ± 0.022 | 0.833 ± 0.011 | 0.697 ± 0.023 |
| Civil Comments | 0.596 ± 0.024 | 0.499 ± 0.007 | 0.588 ± 0.052 | 0.536 ± 0.020 | 0.542 ± 0.016 |
| DBPedia-0 | 0.890 ± 0.016 | 0.514 ± 0.026 | 0.880 ± 0.049 | 0.777 ± 0.055 | 0.782 ± 0.044 |
| DBPedia-1 | 0.833 ± 0.018 | 0.522 ± 0.023 | 0.834 ± 0.061 | 0.768 ± 0.024 | 0.760 ± 0.022 |
| DBPedia-2 | 0.912 ± 0.010 | 0.522 ± 0.031 | 0.916 ± 0.033 | 0.903 ± 0.012 | 0.859 ± 0.019 |
| DBPedia-3 | 0.879 ± 0.021 | 0.520 ± 0.026 | 0.900 ± 0.019 | 0.863 ± 0.018 | 0.812 ± 0.022 |
| IMDB | 0.541 ± 0.041 | 0.510 ± 0.008 | 0.630 ± 0.017 | 0.575 ± 0.024 | 0.540 ± 0.018 |
| Rotten Tomatoes | 0.576 ± 0.042 | 0.495 ± 0.007 | 0.659 ± 0.064 | 0.568 ± 0.027 | 0.542 ± 0.019 |
| SMS Spam | 0.937 ± 0.017 | 0.856 ± 0.080 | 0.913 ± 0.044 | 0.953 ± 0.014 | 0.877 ± 0.036 |
| SST | 0.602 ± 0.015 | 0.509 ± 0.013 | 0.705 ± 0.021 | 0.605 ± 0.043 | 0.542 ± 0.003 |
| Youtube | 0.862 ± 0.045 | 0.626 ± 0.081 | 0.874 ± 0.046 | 0.762 ± 0.052 | 0.654 ± 0.050 |

Table 11: **Standard deviation of accuracy, number of examples = 32, PYTHIA (160M)**

| Dataset | TART | In-context learning | Fine-tuning full | Fine-tuning layer | Adapters |
|---|---|---|---|---|---|
| AG-News-0 | 0.825 ± 0.031 | 0.503 ± 0.003 | 0.802 ± 0.041 | 0.818 ± 0.014 | 0.739 ± 0.049 |
| AG-News-1 | 0.880 ± 0.012 | 0.516 ± 0.026 | 0.898 ± 0.043 | 0.912 ± 0.034 | 0.829 ± 0.010 |
| AG-News-2 | 0.771 ± 0.009 | 0.519 ± 0.025 | 0.736 ± 0.071 | 0.773 ± 0.039 | 0.684 ± 0.040 |
| AG-News-3 | 0.810 ± 0.016 | 0.509 ± 0.013 | 0.753 ± 0.058 | 0.805 ± 0.031 | 0.742 ± 0.030 |
| Civil Comments | 0.587 ± 0.025 | 0.500 ± 0.011 | 0.549 ± 0.016 | 0.564 ± 0.062 | 0.555 ± 0.009 |
| DBPedia-0 | 0.857 ± 0.035 | 0.592 ± 0.083 | 0.801 ± 0.125 | 0.822 ± 0.043 | 0.834 ± 0.038 |
| DBPedia-1 | 0.802 ± 0.031 | 0.558 ± 0.047 | 0.870 ± 0.037 | 0.800 ± 0.041 | 0.813 ± 0.034 |
| DBPedia-2 | 0.879 ± 0.018 | 0.609 ± 0.025 | 0.938 ± 0.018 | 0.920 ± 0.031 | 0.903 ± 0.021 |
| DBPedia-3 | 0.866 ± 0.035 | 0.634 ± 0.110 | 0.812 ± 0.163 | 0.911 ± 0.004 | 0.876 ± 0.037 |
| IMDB | 0.553 ± 0.046 | 0.541 ± 0.024 | 0.636 ± 0.016 | 0.600 ± 0.018 | 0.536 ± 0.024 |
| Rotten Tomatoes | 0.589 ± 0.037 | 0.575 ± 0.039 | 0.713 ± 0.029 | 0.630 ± 0.026 | 0.527 ± 0.015 |
| SMS Spam | 0.933 ± 0.017 | 0.762 ± 0.033 | 0.956 ± 0.026 | 0.905 ± 0.056 | 0.917 ± 0.017 |
| SST | 0.579 ± 0.032 | 0.502 ± 0.020 | 0.739 ± 0.013 | 0.638 ± 0.035 | 0.562 ± 0.017 |
| Youtube | 0.799 ± 0.117 | 0.710 ± 0.182 | 0.887 ± 0.033 | 0.756 ± 0.115 | 0.850 ± 0.024 |

Table 12: **Standard deviation of accuracy, number of examples = 32, BLOOM (560M)**

| Dataset | TART | In-context learning | Fine-tuning full | Fine-tuning layer | Adapters |
|---|---|---|---|---|---|
| AG-News-0 | $0.812 \pm 0.021$ | $0.560 \pm 0.046$ | $0.832 \pm 0.023$ | $0.876 \pm 0.012$ | $0.833 \pm 0.016$ |
| AG-News-1 | $0.904 \pm 0.009$ | $0.623 \pm 0.051$ | $0.924 \pm 0.026$ | $0.932 \pm 0.007$ | $0.923 \pm 0.009$ |
| AG-News-2 | $0.786 \pm 0.008$ | $0.566 \pm 0.023$ | $0.791 \pm 0.016$ | $0.824 \pm 0.010$ | $0.797 \pm 0.010$ |
| AG-News-3 | $0.822 \pm 0.030$ | $0.533 \pm 0.029$ | $0.800 \pm 0.042$ | $0.840 \pm 0.015$ | $0.825 \pm 0.034$ |
| Civil Comments | $0.591 \pm 0.029$ | $0.497 \pm 0.003$ | $0.568 \pm 0.038$ | $0.554 \pm 0.029$ | $0.614 \pm 0.028$ |
| DBPedia-0 | $0.911 \pm 0.011$ | $0.676 \pm 0.097$ | $0.892 \pm 0.017$ | $0.898 \pm 0.012$ | $0.894 \pm 0.023$ |
| DBPedia-1 | $0.823 \pm 0.025$ | $0.644 \pm 0.115$ | $0.838 \pm 0.058$ | $0.819 \pm 0.018$ | $0.833 \pm 0.026$ |
| DBPedia-2 | $0.902 \pm 0.015$ | $0.622 \pm 0.114$ | $0.931 \pm 0.032$ | $0.940 \pm 0.005$ | $0.913 \pm 0.007$ |
| DBPedia-3 | $0.883 \pm 0.023$ | $0.620 \pm 0.129$ | $0.891 \pm 0.015$ | $0.911 \pm 0.006$ | $0.889 \pm 0.021$ |
| IMDB | $0.557 \pm 0.033$ | $0.522 \pm 0.025$ | $0.641 \pm 0.008$ | $0.564 \pm 0.017$ | $0.577 \pm 0.020$ |
| Rotten Tomatoes | $0.572 \pm 0.014$ | $0.548 \pm 0.045$ | $0.710 \pm 0.010$ | $0.575 \pm 0.048$ | $0.572 \pm 0.029$ |
| SMS Spam | $0.882 \pm 0.044$ | $0.860 \pm 0.036$ | $0.946 \pm 0.019$ | $0.881 \pm 0.013$ | $0.925 \pm 0.010$ |
| SST | $0.570 \pm 0.019$ | $0.563 \pm 0.032$ | $0.705 \pm 0.019$ | $0.575 \pm 0.028$ | $0.542 \pm 0.035$ |
| Youtube | $0.810 \pm 0.039$ | $0.792 \pm 0.081$ | $0.923 \pm 0.014$ | $0.874 \pm 0.029$ | $0.832 \pm 0.053$ |

Table 13: **Standard deviation of accuracy, number of examples = 48, GPT-NEO (125M)**

| Dataset | TART | In-context learning | Fine-tuning full | Fine-tuning layer | Adapters |
|---|---|---|---|---|---|
| AG-News-0 | $0.846 \pm 0.018$ | $0.496 \pm 0.007$ | $0.871 \pm 0.015$ | $0.871 \pm 0.005$ | $0.732 \pm 0.036$ |
| AG-News-1 | $0.923 \pm 0.005$ | $0.528 \pm 0.044$ | $0.942 \pm 0.004$ | $0.935 \pm 0.009$ | $0.780 \pm 0.040$ |
| AG-News-2 | $0.798 \pm 0.009$ | $0.509 \pm 0.004$ | $0.731 \pm 0.040$ | $0.782 \pm 0.018$ | $0.722 \pm 0.019$ |
| AG-News-3 | $0.845 \pm 0.012$ | $0.515 \pm 0.014$ | $0.787 \pm 0.046$ | $0.852 \pm 0.011$ | $0.718 \pm 0.026$ |
| Civil Comments | $0.605 \pm 0.037$ | $0.512 \pm 0.009$ | $0.607 \pm 0.039$ | $0.556 \pm 0.014$ | $0.556 \pm 0.018$ |
| DBPedia-0 | $0.905 \pm 0.020$ | $0.549 \pm 0.057$ | $0.924 \pm 0.020$ | $0.830 \pm 0.034$ | $0.798 \pm 0.021$ |
| DBPedia-1 | $0.840 \pm 0.021$ | $0.592 \pm 0.062$ | $0.867 \pm 0.024$ | $0.808 \pm 0.031$ | $0.780 \pm 0.011$ |
| DBPedia-2 | $0.916 \pm 0.009$ | $0.636 \pm 0.081$ | $0.937 \pm 0.017$ | $0.923 \pm 0.013$ | $0.870 \pm 0.022$ |
| DBPedia-3 | $0.888 \pm 0.022$ | $0.618 \pm 0.096$ | $0.927 \pm 0.011$ | $0.885 \pm 0.007$ | $0.827 \pm 0.026$ |
| IMDB | $0.557 \pm 0.034$ | $0.503 \pm 0.008$ | $0.632 \pm 0.013$ | $0.566 \pm 0.030$ | $0.545 \pm 0.008$ |
| Rotten Tomatoes | $0.601 \pm 0.026$ | $0.480 \pm 0.014$ | $0.683 \pm 0.033$ | $0.556 \pm 0.033$ | $0.562 \pm 0.018$ |
| SMS Spam | $0.956 \pm 0.008$ | $0.869 \pm 0.044$ | $0.966 \pm 0.011$ | $0.960 \pm 0.008$ | $0.891 \pm 0.035$ |
| SST | $0.612 \pm 0.024$ | $0.500 \pm 0.025$ | $0.702 \pm 0.046$ | $0.594 \pm 0.042$ | $0.560 \pm 0.023$ |
| Youtube | $0.872 \pm 0.019$ | $0.654 \pm 0.063$ | $0.904 \pm 0.033$ | $0.826 \pm 0.046$ | $0.666 \pm 0.027$ |

Table 14: **Standard deviation of accuracy, number of examples = 48, PYTHIA (160M)**

| Dataset | TART | In-context learning | Fine-tuning full | Fine-tuning layer | Adapters |
|---|---|---|---|---|---|
| AG-News-0 | $0.829 \pm 0.029$ | $0.507 \pm 0.005$ | $0.829 \pm 0.071$ | $0.847 \pm 0.022$ | $0.788 \pm 0.028$ |
| AG-News-1 | $0.909 \pm 0.010$ | $0.543 \pm 0.052$ | $0.940 \pm 0.011$ | $0.935 \pm 0.004$ | $0.856 \pm 0.014$ |
| AG-News-2 | $0.789 \pm 0.016$ | $0.511 \pm 0.012$ | $0.715 \pm 0.057$ | $0.793 \pm 0.027$ | $0.708 \pm 0.011$ |
| AG-News-3 | $0.832 \pm 0.018$ | $0.530 \pm 0.038$ | $0.834 \pm 0.029$ | $0.843 \pm 0.014$ | $0.763 \pm 0.015$ |
| Civil Comments | $0.602 \pm 0.030$ | $0.506 \pm 0.011$ | $0.605 \pm 0.044$ | $0.592 \pm 0.039$ | $0.566 \pm 0.008$ |
| DBPedia-0 | $0.889 \pm 0.022$ | $0.703 \pm 0.085$ | $0.933 \pm 0.021$ | $0.888 \pm 0.024$ | $0.873 \pm 0.027$ |
| DBPedia-1 | $0.817 \pm 0.023$ | $0.734 \pm 0.076$ | $0.891 \pm 0.028$ | $0.836 \pm 0.036$ | $0.834 \pm 0.024$ |
| DBPedia-2 | $0.900 \pm 0.011$ | $0.847 \pm 0.037$ | $0.949 \pm 0.013$ | $0.940 \pm 0.009$ | $0.914 \pm 0.013$ |
| DBPedia-3 | $0.884 \pm 0.020$ | $0.815 \pm 0.105$ | $0.928 \pm 0.024$ | $0.917 \pm 0.015$ | $0.891 \pm 0.037$ |
| IMDB | $0.565 \pm 0.033$ | $0.545 \pm 0.033$ | $0.641 \pm 0.020$ | $0.604 \pm 0.022$ | $0.542 \pm 0.015$ |
| Rotten Tomatoes | $0.605 \pm 0.015$ | $0.505 \pm 0.005$ | $0.704 \pm 0.025$ | $0.672 \pm 0.042$ | $0.531 \pm 0.018$ |
| SMS Spam | $0.933 \pm 0.009$ | $0.636 \pm 0.130$ | $0.929 \pm 0.058$ | $0.919 \pm 0.032$ | $0.914 \pm 0.027$ |
| SST | $0.610 \pm 0.030$ | $0.489 \pm 0.004$ | $0.695 \pm 0.020$ | $0.673 \pm 0.029$ | $0.538 \pm 0.011$ |
| Youtube | $0.834 \pm 0.049$ | $0.797 \pm 0.090$ | $0.805 \pm 0.095$ | $0.851 \pm 0.020$ | $0.865 \pm 0.012$ |

Table 15: **Standard deviation of accuracy, number of examples = 48, BLOOM (560M)**

| Dataset | TART | In-context learning | Fine-tuning full | Fine-tuning layer | Adapters |
|---|---|---|---|---|---|
| AG-News-0 | $0.817 \pm 0.024$ | $0.539 \pm 0.044$ | $0.855 \pm 0.030$ | $0.877 \pm 0.013$ | $0.830 \pm 0.023$ |
| AG-News-1 | $0.904 \pm 0.007$ | $0.561 \pm 0.048$ | $0.923 \pm 0.027$ | $0.939 \pm 0.003$ | $0.922 \pm 0.010$ |
| AG-News-2 | $0.790 \pm 0.026$ | $0.576 \pm 0.022$ | $0.814 \pm 0.009$ | $0.839 \pm 0.009$ | $0.816 \pm 0.014$ |
| AG-News-3 | $0.833 \pm 0.015$ | $0.550 \pm 0.035$ | $0.803 \pm 0.017$ | $0.852 \pm 0.013$ | $0.842 \pm 0.018$ |
| Civil Comments | $0.581 \pm 0.039$ | $0.499 \pm 0.002$ | $0.587 \pm 0.018$ | $0.576 \pm 0.039$ | $0.605 \pm 0.036$ |
| DBPedia-0 | $0.915 \pm 0.005$ | $0.652 \pm 0.095$ | $0.917 \pm 0.011$ | $0.908 \pm 0.019$ | $0.909 \pm 0.025$ |
| DBPedia-1 | $0.825 \pm 0.037$ | $0.633 \pm 0.104$ | $0.838 \pm 0.032$ | $0.829 \pm 0.012$ | $0.852 \pm 0.011$ |
| DBPedia-2 | $0.887 \pm 0.020$ | $0.606 \pm 0.088$ | $0.950 \pm 0.007$ | $0.952 \pm 0.011$ | $0.916 \pm 0.014$ |
| DBPedia-3 | $0.873 \pm 0.033$ | $0.611 \pm 0.136$ | $0.898 \pm 0.034$ | $0.927 \pm 0.007$ | $0.909 \pm 0.008$ |
| IMDB | $0.558 \pm 0.029$ | $0.515 \pm 0.024$ | $0.646 \pm 0.010$ | $0.563 \pm 0.021$ | $0.566 \pm 0.035$ |
| Rotten Tomatoes | $0.601 \pm 0.020$ | $0.533 \pm 0.053$ | $0.708 \pm 0.015$ | $0.579 \pm 0.049$ | $0.556 \pm 0.033$ |
| SMS Spam | $0.869 \pm 0.023$ | $0.825 \pm 0.074$ | $0.934 \pm 0.016$ | $0.898 \pm 0.008$ | $0.927 \pm 0.004$ |
| SST | $0.570 \pm 0.038$ | $0.554 \pm 0.041$ | $0.712 \pm 0.033$ | $0.609 \pm 0.021$ | $0.567 \pm 0.016$ |
| Youtube | $0.790 \pm 0.050$ | $0.819 \pm 0.050$ | $0.926 \pm 0.014$ | $0.891 \pm 0.031$ | $0.837 \pm 0.055$ |

Table 16: **Standard deviation of accuracy, number of examples = 64, GPT-NEO (125M)**

| Dataset | TART | In-context learning | Fine-tuning full | Fine-tuning layer | Adapters |
|---|---|---|---|---|---|
| AG-News-0 | $0.851 \pm 0.024$ | $0.502 \pm 0.003$ | $0.858 \pm 0.009$ | $0.869 \pm 0.010$ | $0.764 \pm 0.030$ |
| AG-News-1 | $0.925 \pm 0.002$ | $0.529 \pm 0.030$ | $0.937 \pm 0.011$ | $0.938 \pm 0.004$ | $0.820 \pm 0.021$ |
| AG-News-2 | $0.812 \pm 0.007$ | $0.513 \pm 0.013$ | $0.752 \pm 0.051$ | $0.795 \pm 0.010$ | $0.738 \pm 0.008$ |
| AG-News-3 | $0.851 \pm 0.007$ | $0.503 \pm 0.004$ | $0.820 \pm 0.020$ | $0.855 \pm 0.009$ | $0.741 \pm 0.019$ |
| Civil Comments | $0.606 \pm 0.029$ | $0.500 \pm 0.001$ | $0.659 \pm 0.032$ | $0.566 \pm 0.023$ | $0.566 \pm 0.018$ |
| DBPedia-0 | $0.910 \pm 0.013$ | $0.518 \pm 0.020$ | $0.912 \pm 0.027$ | $0.858 \pm 0.019$ | $0.825 \pm 0.015$ |
| DBPedia-1 | $0.839 \pm 0.027$ | $0.542 \pm 0.028$ | $0.897 \pm 0.016$ | $0.824 \pm 0.031$ | $0.788 \pm 0.018$ |
| DBPedia-2 | $0.916 \pm 0.011$ | $0.609 \pm 0.106$ | $0.953 \pm 0.013$ | $0.922 \pm 0.012$ | $0.882 \pm 0.019$ |
| DBPedia-3 | $0.887 \pm 0.028$ | $0.527 \pm 0.022$ | $0.940 \pm 0.013$ | $0.904 \pm 0.007$ | $0.857 \pm 0.020$ |
| IMDB | $0.556 \pm 0.024$ | $0.506 \pm 0.005$ | $0.619 \pm 0.030$ | $0.574 \pm 0.017$ | $0.552 \pm 0.009$ |
| Rotten Tomatoes | $0.624 \pm 0.024$ | $0.485 \pm 0.020$ | $0.686 \pm 0.040$ | $0.577 \pm 0.033$ | $0.568 \pm 0.019$ |
| SMS Spam | $0.937 \pm 0.018$ | $0.905 \pm 0.017$ | $0.960 \pm 0.021$ | $0.961 \pm 0.006$ | $0.899 \pm 0.014$ |
| SST | $0.606 \pm 0.022$ | $0.508 \pm 0.022$ | $0.688 \pm 0.047$ | $0.602 \pm 0.036$ | $0.567 \pm 0.017$ |
| Youtube | $0.888 \pm 0.028$ | $0.715 \pm 0.097$ | $0.897 \pm 0.046$ | $0.878 \pm 0.050$ | $0.675 \pm 0.019$ |

Table 17: **Standard deviation of accuracy, number of examples = 64, PYTHIA (160M)**

| Dataset | TART | In-context learning | Fine-tuning full | Fine-tuning layer | Adapters |
|---|---|---|---|---|---|
| AG-News-0 | $0.836 \pm 0.018$ | $0.509 \pm 0.008$ | $0.850 \pm 0.027$ | $0.856 \pm 0.008$ | $0.799 \pm 0.029$ |
| AG-News-1 | $0.918 \pm 0.007$ | $0.543 \pm 0.033$ | $0.900 \pm 0.024$ | $0.933 \pm 0.008$ | $0.875 \pm 0.015$ |
| AG-News-2 | $0.799 \pm 0.012$ | $0.515 \pm 0.019$ | $0.784 \pm 0.018$ | $0.831 \pm 0.022$ | $0.732 \pm 0.017$ |
| AG-News-3 | $0.836 \pm 0.011$ | $0.504 \pm 0.003$ | $0.811 \pm 0.034$ | $0.853 \pm 0.011$ | $0.784 \pm 0.022$ |
| Civil Comments | $0.602 \pm 0.030$ | $0.510 \pm 0.012$ | $0.573 \pm 0.035$ | $0.611 \pm 0.025$ | $0.572 \pm 0.013$ |
| DBPedia-0 | $0.905 \pm 0.015$ | $0.667 \pm 0.052$ | $0.936 \pm 0.018$ | $0.902 \pm 0.022$ | $0.882 \pm 0.020$ |
| DBPedia-1 | $0.809 \pm 0.022$ | $0.687 \pm 0.117$ | $0.887 \pm 0.039$ | $0.852 \pm 0.032$ | $0.853 \pm 0.008$ |
| DBPedia-2 | $0.881 \pm 0.026$ | $0.799 \pm 0.075$ | $0.955 \pm 0.022$ | $0.947 \pm 0.009$ | $0.922 \pm 0.014$ |
| DBPedia-3 | $0.877 \pm 0.014$ | $0.793 \pm 0.106$ | $0.906 \pm 0.027$ | $0.921 \pm 0.017$ | $0.899 \pm 0.024$ |
| IMDB | $0.571 \pm 0.033$ | $0.539 \pm 0.020$ | $0.621 \pm 0.039$ | $0.618 \pm 0.013$ | $0.542 \pm 0.012$ |
| Rotten Tomatoes | $0.597 \pm 0.025$ | $0.536 \pm 0.047$ | $0.684 \pm 0.053$ | $0.672 \pm 0.041$ | $0.543 \pm 0.024$ |
| SMS Spam | $0.907 \pm 0.045$ | $0.659 \pm 0.133$ | $0.942 \pm 0.024$ | $0.931 \pm 0.030$ | $0.909 \pm 0.033$ |
| SST | $0.620 \pm 0.039$ | $0.495 \pm 0.015$ | $0.672 \pm 0.039$ | $0.716 \pm 0.032$ | $0.554 \pm 0.020$ |
| Youtube | $0.844 \pm 0.059$ | $0.765 \pm 0.137$ | $0.879 \pm 0.058$ | $0.865 \pm 0.033$ | $0.883 \pm 0.016$ |

Table 18: **Standard deviation of accuracy, number of examples = 64, BLOOM (560M)**

