# OpenReview forum: "TART: A plug-and-play Transformer module for task-agnostic reasoning"
_NeurIPS.cc/2023/Conference — NeurIPS 2023 poster_

### Official Review · Reviewer_CKNq · 2023-07-05

**Soundness:** 3 good
**Presentation:** 4 excellent
**Contribution:** 3 good
**Rating:** 6
**Confidence:** 4

**Summary:**

This paper proposes and evaluates a method (TART) for improving the in-context learning of large, pretrained base models applied to downstream binary classification tasks. TART stacks a second transformer on top of the pretrained base transformer. The TART transformer is trained to rely on in-context examples to solve a large number of synthetically generated binary classification problems. TART’s training is task-agnostic in the sense that its synthetic training data is unrelated to the domain of the downstream tasks on which the combined base+TART model is evaluated. The experiments demonstrate that TART improves the combined model’s in-context learning across a range of binary classification tasks, and a diverse set of base models pretrained on language, vision, and audio data.

**Strengths:**

Given the profound importance of large pretrained transformer models today, and their remarkable ability to learn from in-context examples, general methods for further boosting the performance of in-context learning are of great interest.

The paper’s proposed method, TART, is a well-motivated and novel approach. And the paper’s experiments clearly demonstrate the degree to which TART boosts the in-context learning of large, pretrained base models applied to downstream binary classification tasks. Exciting results!

Finally, as noted in section 5 of the paper, the TART approach may lead to similar methods that improve LLM performance on a wider range of tasks beyond binary classification.

**Weaknesses:**

The paper claims to make three main contributions, which I summarize here:

1.	WHAT CAUSES THE ICL GAP?  Study why in-context learning does not perform as well as task-specific fine-tuning despite having access to the same information, via a representation-reasoning decomposition.

2.	TART CLOSES THE ICL GAP  Propose a new task-agnostic method, TART, which bridges the performance gap to task-specific methods and is trained using only synthetic data.

3.	TART IS GENERAL  Demonstrate that TART works across different NLP tasks for a range of model families. The same inference module generalizes to vision and speech domains as well.

I follow this outline of claims in describing what I see as the paper’s weaknesses.

Claim 1. WHAT CAUSES THE ICL GAP?

The paper mounts an ambitious study of the gap in performance between in-context learning and task-specific fine-tuning, given a common set of examples. The experimental results are suggestive, providing ample motivation for the TART approach. However, the paper ventures far beyond the question of TART’s motivation when it attempts to draw broad conclusions about a clear representation-reasoning decomposition. The field has arrived at no settled definition of reasoning, so any distinction between representation and reasoning remains nebulous. Transformers themselves illustrate the complex, intertwined relationship between representation and reasoning.

As a result the paper makes extraordinary claims (requiring extraordinary evidence) when it equates reasoning with probabilistic inference, and when it makes statements like the following:

•	“LLMs lack reasoning abilities”

•	“LLMs lack the ability to perform simple reasoning over their learned representations”

•	“this performance gap exists due to their inability to perform simple probabilistic reasoning tasks.”

I believe that too much of the paper is devoted to pursuing such a challenging case, which would require far more evidence than the experiments provide. And I see it as unnecessary, since the experimental results are more than sufficient to provide adequate motivation for the TART approach. Furthermore, the paper’s heavy emphasis on its reasoning-centric terminology (like “reasoning module” or “probabilistic inference module”) obscures the relatively simple and limited nature of TART. To be concrete, the paper doesn’t need to wade into the philosophy of reasoning vs. representation to make the case that TART (as one might expect) boosts in-context learning by training a transformer (on task-agnostic, synthetic data!) to perform well at in-context learning.

In summary, while I agree that the paper does cast some useful light on the performance gap between in-context learning and task-specific fine-tuning, much mystery remains there, so I don’t see this as a strong contribution of the paper, although it is sufficient to motivate the approach.

Claim 2. TART CLOSES THE ICL GAP

The experiments demonstrate to me that TART does succeed in largely closing the in-context learning gap, at least when given enough examples. But what happens when only a few examples are available at test time? The paper presents no performance comparisons against baselines tested on fewer than 20 examples, as far as I can see. Since the TART transformer is a task-agnostic module sitting on top of the base model, one would expect its performance to be no better than random in the absence of task-specific examples at test time. So how many test-time examples does it take for TART to show good results? Below 20 examples is an important range to explore, since in-context learning is known to be able to derive great benefit from as few as one or two examples. If it turns out that TART requires at least, for example, 10 examples to deliver benefit, then the paper would be over-claiming when it says that TART, “when composed with any LLM via its embeddings, generically improves upon its reasoning abilities.”

The all-important technical details of TART’s training are poorly explained in the current version of the paper. Section 3.2.1 leaves too much to the reader’s imagination. Line 217 talks about updating the parameters $w$, which must therefore refer to the parameters of the TART transformer. But equation (3) also contains a symbol $w_t$ which cannot refer to the transformer’s parameters. Rather, the $w_t$ must refer to the randomly sampled weights for the linear layer used to define a binary classification problem that the transformer is trained on. At least this is my current understanding. The apparent conflict of notation needs to be sorted out, and eq. (3) should be explained much more carefully in the text. For instance, it’s worth mentioning that the angle brackets refer to a dot product, since this notation is rarely used in papers on transformers.

The paper would be easier to follow if it used the term logistic regression correctly. Specifically, logistic regression is not a problem or a task. Rather, logistic regression is a method that can be used in solving classification problems. The term is used correctly on the following lines:

Line 231  “For each problem, we train task-specific linear classifiers using logistic regression”

Line 535  “To conduct linear probing over the embeddings, we perform logistic regression over the output embeddings of each model and the given labels in the training set using the built-in logistic regression solver from the scikit-learn python library, utilizing the lbgfs solver.”

But the rest of the paper uses the term “logistic regression problem” as if logistic regression were a problem type rather than a solution method.

Terminology for eq. (3) is tricky, because it involves (I believe) a linear layer of random weights which could be (but are not) obtained by the logistic regression method. And these random weights are used, along with random covariates $x$, to generate synthetic classification problems. Because the structure of this generator is related to the method of logistic regression, it is tempting to refer to the generated classification problems as logistic regression problems. But that’s conflating problems with methods. I suggest spending a few sentences to explain eq. (3) in simple terms, and replacing the phrase *logistic regression problem/task* with *binary classification problem/task* throughout the paper.

It seems that positional encodings must be required in training the TART transformer, to establish the alternating positions of $x$ and $y$ vectors in the sequences. But I find no mention of positional encodings in the paper.


Claim 3. TART IS GENERAL

It is shown that TART works across different binary classification tasks and model families, including vision and speech domains. However, the results are all limited to binary classification, which is not a very general class. As explained in the discussion, “In future work, we seek to understand whether synthetic tasks exist for training other generic reasoning modules, capable of improving base LLM performance on tasks such as generation or summarization.” Such results would be highly significant. Until then, the current restriction to binary classification limits the significance of this contribution.


MINOR SUGGESTIONS

Line 130  “at par when compared with” should be changed to “on par with”.

Line 191  “TART comprises of two components” should be changed to either “TART comprises two components” or “TART is comprised of two components”.

Line 584   A transformer outputs a sequence of vectors, not embeddings.


CONCLUSION

Because of the problems noted in this section, I would vote to reject the paper in its current form. However, I have confidence that the authors can address all of these issues in the next version of the paper, so for now I’m optimistically voting for acceptance.

**Questions:**

I assume that TART stands for Task-Agnostic Reasoning Transformer. Is this explained somewhere?

Line 589  If the context length is 258, and each example is one pair, then there would be 129 in-context examples, not 256. Right?

I’m not sure what’s meant by “the binary labels y are encoded as a one-hot vector to match the input dimension $d$ of the corresponding covariates $x$.” Does this mean that only 2 of the 16 dimensions are ever 1.0, while the other 14 dimensions are always 0.0?

**Limitations:**

No problems noted.

---

> ### Author Rebuttal · Authors · 2023-08-10
>
> Thank you for the detailed comments and helpful feedback on our work. We are glad you found our approach well-motivated, novel and exciting.
>
> We completely agree with the three main contributions that you highlighted in your review. While we addressed some of these concerns in the general response (study on representation-reasoning trade-off, generalization to multiclass, and TART in low-data regime), we provide a detailed response below.
>
> **Reasoning vs representation.** We completely agree that there is no consensus in the field on a definition of a reasoning vs. representation, and much less so for in-context learning which is quite nascent. As we mentioned in the general response, this motivated us to make an attempt at explaining why ICL fails. We make a first step towards this understanding through our representation-reasoning decomposition which we mathematically characterize in Equation (1). We feel that this is an important and interesting contribution in and of itself, and while just a first step, we believe it will indeed be helpful and inspiring in spurring further research along these directions.
>
> Our representation-reasoning definitions as well as the analysis with Hypotheses H1-H3 is currently focussed on binary classification tasks in this draft. Our statements and claims are indeed supposed to be restricted to this setup. Following our new extensions to multiclass tasks (see attached PDF for experimental results), we plan to extend this analysis to the multiclass tasks as well.
>
> **TART in low in-context example regime.** For most plots in our paper, we study the performance of TART (and baselines) beginning with 16 in-context examples (k). In the attached pdf, we show that the TART works even in the extremely low-data regime (with k = 4, 6, 8, 10). TART can improve the base ICL performance by up to 22.2% (k=4), 30.1% (k=6), 24.1% (k=8) and 31.3% (k=10). See Table 3 in attached PDF.
>
> Furthermore, on 2 out of 3 datasets, we see that TART outperforms full model finetuning by up to 15.8%. While it is challenging to adapt the LLM for the task with such few data points, TART is able to better use the sample information. We will update our draft with a more extensive study across all benchmarks in this regime.
>
> **Beyond binary classification.** Thank you for this suggestion. In response to this feedback, we train a multiclass TART module which outperforms the ICL baseline by over 59 points (see attached PDF for more details). We realized that our training algorithm was not specific to binary classification tasks and could be easily extended to multiclass problems by using the multinomial logistic regression as the synthetic. We will update the draft to feature these extensions in Sec. 4 (Experimental evaluation).
>
> **Use of term logistic regression.** We have actually used the term logistic regression in an overloaded manner to refer to both the logistic regression generative model as well as the estimator obtained by solving the ERM with respect to this model. We will clarify this in the paper and make it clear which of these we are referring to at a given time. Note that the logistic regression model is a very specific generative model of binary classification which we use in our paper for the synthetics.
>
> **Minor Comments.** We address the some of the minor concerns here:
> - TART stands for Task-Agnostic Reasoning Transformer (mentioned in Sec. 3 header). We agree that the acronym could be explained earlier and we will reflect this change in our draft.
> - For context length = 258, the number of examples should be 129. We will correct this in the updated draft.
> - For the binary classification task, we only use the last dimension out of the 16 to set the labels – it is 1 when the label is positive and 0 when the label is negative. Such a mapping is quite common and has been previously used in [1].
>
> *References*
>
> [1]  Garg, Shivam, et al. "What can transformers learn in-context? a case study of simple function classes." Advances in Neural Information Processing Systems (2022)

---

> > ### Comment · Reviewer_CKNq · 2023-08-18
> > **Response to rebuttal**
> >
> > I applaud the authors for their detailed rebuttals, and especially for the additional experiments which addressed a couple of my concerns (TART when given few examples, and multiclass tasks). Unfortunately, the authors did not even mention my major points regarding the lack of clarity in describing the central method:  The terse coverage of section 3.2.1 and especially Eq. (3), the apparent dual usage of the symbol $w$, and the key question of positional encodings. My initial rating was based on an optimistic assumption that a revised version of the writeup would be attached. Without reading it, I have little confidence that the published paper would explain the method well enough to benefit the community. For this reason, I am lowering my score from 5 to 4.

---

> > > ### Author Response · Authors · 2023-08-18
> > >
> > > Thanks a lot for taking out time to read our rebuttal.
> > >
> > > Unfortunately, the conference does not allow us to submit a revised version of the draft and we are not allowed to put in any text in the attached pdf. However, we do provide a concrete retelling of the story in the global response which will allow us to restructure and add more content to our revised draft.
> > >
> > > - We added the clarification to the dual usage of the symbol $w$ in response to Reviewer hArx (https://openreview.net/forum?id=ZXbgVm3PSt&noteId=IdKRUWCfRs). It is a minor typo and should instead be $\theta$.
> > > - We will pull up the details of the algorithm from Appendix C.1 to section 3 -- it is already there and simply needs to be accommodated with the existing content.
> > > - Yes, the default GPT-2 architecture that we use (see Figure 15 in the draft) has absolute positional encoding for each position. We will clarify this in more detail in the paper.
> > >
> > > We hope this addresses some of the concerns you mentioned above.

---

> > > > ### Comment · Reviewer_CKNq · 2023-08-18
> > > > **Response**
> > > >
> > > > I appreciate the quick clarification, which helps. But Appendix C.1 says nothing that would help readers digest the austere equation (3). There needs to be a clear and detailed explanation of how the task-agnostic training dataset (used to train the TART module) is constructed. These details might even belong in the appendix, hopefully including a few sample examples for illustration. These missing details are at the core of the technique and arguably the most important part of the paper. Did I somehow miss them?

---

> > > > > ### Author Response · Authors · 2023-08-19
> > > > > **clarifying eq (3)**
> > > > >
> > > > > We really appreciate you taking the time for this discussion and your quick response. I think we misinterpreted your point in the review; please allow us to clarify your doubt.
> > > > >
> > > > > Equation (3) is the fundamental logit model (linear logistic model) which describes how we sample the synthetic data which lies in a d-dimensional space. This statistical model has been the basis for modeling uncertainty with discrete-valued output variables, since as early as 1940s [1].
> > > > >
> > > > > In more detail, the sampling process goes as follows:
> > > > > - the regressor $w_t$ is sample from a standard Normal distribution
> > > > > - the features $x_{i,t}$ are sampled from a standard Normal distribution
> > > > > - the corresponding $y_{i,t}$ are sampled based on the sigmoid of the inner product of $x_t$ with $w_t$
> > > > >
> > > > > Figure 2 (left) gives a demonstration of a 2d dataset sampled from the process -- it is simply drawing noisily linearly separable data in d-dimensions. The important point to note here is that it has nothing to do with downstream natural language tasks, and hence agnostic to them. High-dimensional variants of these correspond to datapoints on a d-dimensional sphere with noisy labels which is almost-linearly separable. Given this generation process, Figure 15 describes how we place these sampled real-valued data points in the TART architecture.
> > > > >
> > > > > Additionally, if you see Appendix A.1, Equation (5): this is equivalent to Equation (3) in the natural language setup. Lines 508-511 actually provide concrete examples of how such sequences look like when projected onto the language domain. This subsection contains more intuition and details into the sampling procedure.
> > > > >
> > > > > We understand your concern that readers might not be familiar with this logit models, and we will add in additional discussion describing the process above as background.
> > > > >
> > > > > [1] Cramer, J. S. (2004). "The early origins of the logit model". Studies in History and Philosophy of Science Part C: Studies in History and Philosophy of Biological and Biomedical Sciences.

---

> > > > > > ### Comment · Reviewer_CKNq · 2023-08-19
> > > > > > **Response**
> > > > > >
> > > > > > It's not a question of whether readers are familiar with logit models, especially when the paper doesn't explicitly mention the *logit model* at all. (And as we have discussed, the paper's usage of the term *logistic regression* muddies the waters, raising suspicion regarding the paper's usage of technical terms in general.) It's a question of whether the paper explains how the task-agnostic data is generated with sufficient clarity for readers to feel confident that they understand and could apply the technique themselves. I believe the bullet points in your last comment provide the right level of detail for walking readers through eq. (3). The reference to Fig. 2 is also helpful as an illustration. No need to get pedantic though, as when citing the early history of the logit model.

---

> > > > > > > ### Author Response · Authors · 2023-08-19
> > > > > > >
> > > > > > > Thank you for your suggestion and time. We will add the bullets above as an explanation right after Eq. (3) and highlight the reference to Figure 2 there. The logit model is simply another name for the logistic model which is often used interchangeably -- we will be sure to be consistent in our usage in the updated manuscript.
> > > > > > >
> > > > > > > We hope that this addresses all your concerns. We will make sure to make these important modifications to our draft; these suggestions will help improve the presentation of our work.
> > > > > > >
> > > > > > > If you have any other questions, please do let us know and we will try our best to address it. We really appreciate you taking the time to discuss.

---

> > > > > > > > ### Comment · Reviewer_CKNq · 2023-08-19
> > > > > > > > **Response**
> > > > > > > >
> > > > > > > > Of course you're right that *logit model* and *logistic model* are the same thing, but the paper uses neither term. The paper uses the term *logistic regression*, which is not a model at all, but a method for estimating the parameters of a logistic model. If the paper confuses *logistic regression* with *logistic model*, many readers will be confused as well.
> > > > > > > >
> > > > > > > > Despite my lingering concern that the paper will not adequately handle these technical explanations, I'm impressed by the extra experiments provided in rebuttal, so I'll raise my score back to its original value.

---

> > > > > > > > > ### Author Response · Authors · 2023-08-19
> > > > > > > > > **Changes to draft**
> > > > > > > > >
> > > > > > > > > Thank you so much for engaging in this constructive discussion. We agree that the paper draft, in its current form, has some issues with clarity which you have raised through your review and discussion. While we are constrained by the conference rules to not upload an updated pdf, we would like to highlight the changes we have made to our revised draft, to make sure we adequately address your concerns.
> > > > > > > > >
> > > > > > > > > > Clarifying logistic model vs logistic regression
> > > > > > > > >
> > > > > > > > > Thanks for bringing up this distinction. We completely agree that this overuse of the term logistic regression problem can lead to confusion for the reader. Based on your feedback, we are changing the following lines.
> > > > > > > > > - Line 61: synthetically generated tasks from the logistic (or logit) model
> > > > > > > > > - Figure 2: synthetically generated tasks from the logistic model
> > > > > > > > > - Line 215: input sequence corresponding to  a different problem sampled from the logistic model
> > > > > > > > > - Line 218: parameters $\theta$ comprises a different d-dimensional problem sampled from the logistic model given by:
> > > > > > > > > - Line 230: 64 different problems sampled from the logistic model
> > > > > > > > > - Line 238: noise level, $\alpha$, of the logistic model in Eq. (3)
> > > > > > > > > - Line 273: generalization error on the tasks sampled from the logistic model
> > > > > > > > >
> > > > > > > > > > Understanding Eq. (3)
> > > > > > > > >
> > > > > > > > > Lines 219-220 have been updated to:
> > > > > > > > > where $\sigma$ represents the sigmoid function and the multiplier $\alpha$ determines the noise level of the problem. The above sampling model comprises three components, the regressor $w_t$, features $x_{i,t}$ and labels $y_{i,t}$ sampled as
> > > > > > > > > - the regressor $w_t$ is sample from a standard Normal distribution
> > > > > > > > > - the features $x_{i,t}$ are sampled from a standard Normal distribution
> > > > > > > > > - the corresponding $y_{i,t}$ are sampled based on the sigmoid of the inner product of $x_t$ with $w_t$.
> > > > > > > > >
> > > > > > > > > See Figure 2 (left) for a demonstration of a 2d dataset sampled from this process -- it is simply drawing noisy linearly separable data in d-dimensions.
> > > > > > > > >
> > > > > > > > > > Confusion regarding symbol $w$
> > > > > > > > >
> > > > > > > > > Line 217 is updated to "Each training sequence $s_t$ used to update the parameters $\theta$"
> > > > > > > > >
> > > > > > > > > > Architecture details
> > > > > > > > >
> > > > > > > > > We have updated line 207 to: The GPT-2 architecture comprises a causal (unidirectional) transformer with 12 decoder layers and 8 attention heads per layer for a total of 22 M parameters. The model has absolute positional encodings for each position. Please see Appendix C.1 for more details.
> > > > > > > > >
> > > > > > > > > Additionally we will also add the model card describing the type of attention (local vs global) to our appendix.
> > > > > > > > >
> > > > > > > > > > Additional experiments on low-data and multi-class settings
> > > > > > > > >
> > > > > > > > > As shown in the attached pdf, we have added two additional sub-sections in the evaluation section (Sec. 4). Section 4.2 has been updated with an additional paragraph on performance in low-data regime. An additional section has been added with extension to Multi-class tasks (Sec 4.4).
> > > > > > > > >
> > > > > > > > > > Reasoning vs. representation
> > > > > > > > >
> > > > > > > > > To make sure our claims are within the scope, we have updated:
> > > > > > > > > - Line 108: understand their relative performance for binary classification tasks.
> > > > > > > > > - Line 111: downstream binary classification tasks.
> > > > > > > > > - Line 144: This section investigates why this performance gap exists for binary classification tasks.
> > > > > > > > > - Line 151: insufficient reasoning abilities, i.e., the ability to learn linear decision boundary for a binary classification task
> > > > > > > > > - The hypothesis H1-H3 have been updated to focus on binary classification tasks.
> > > > > > > > > - Additionally, we will add an extra section in the appendix focussing on the same analysis with multi-class tasks.
> > > > > > > > >
> > > > > > > > > > Minor comments
> > > > > > > > >
> > > > > > > > > - The acronym for TART has been added to Line 59: We propose TART (Task Agnostic Reasoning Transformer)
> > > > > > > > > - Line 191 has been corrected to "TART comprises two components"
> > > > > > > > > - Line 589 has been updated to 129 in-context examples
> > > > > > > > >
> > > > > > > > > ---
> > > > > > > > >
> > > > > > > > > Thank you so much for helping us improve the readability and clarity of our manuscript. We sincerely hope these changes and the retelling of our story from the global response address your concerns and that our revised manuscript is up to the high standards you expect.

---

> > > > > > > > > > ### Comment · Reviewer_CKNq · 2023-08-19
> > > > > > > > > > **Response**
> > > > > > > > > >
> > > > > > > > > > Yes, these are great improvements to the presentation! I've raised my score one notch.

---

> > > > > > > > > > > ### Author Response · Authors · 2023-08-19
> > > > > > > > > > > **Thanks**
> > > > > > > > > > >
> > > > > > > > > > > Thank you for your thoughtful comments and feedback. The review and discussion has gone a long way in improving our paper's presentation and clarity. We really appreciate your time and effort!

---

### Official Review · Reviewer_tMpn · 2023-07-07

**Soundness:** 4 excellent
**Presentation:** 4 excellent
**Contribution:** 4 excellent
**Rating:** 8
**Confidence:** 4

**Summary:**

The authors study why in context learning doesn't perform as well as task specific fine tuning by decomposing the process of in context learning into representation and reasoning. They discover that the gap is due mostly to deficiencies in reasoning and propose a new method called TART to bridge the gap. They demonstrate that TART is task and model agnostic.

**Strengths:**

Originality:
The authors note and address the problem that in-context learning underperforms model fine tuning. The paper's primary contributions are disentangling representation strength and reasoning ability and finding a way to measure them, then using the insight from their measurements to improve the model's reasoning ability and thus close the gap between in context learning and fine tuning. Both are very original.
I thought the concept of training a reasoning model to do probabilistic inference using synthetic data sets was clever. The same for averaging embeddings of multi-token data points in order to piece together the base model and the reasoning module.
I also like thinking of LLM reasoning as probablistic inference in the first place. That seems insightful.

Quality and Clarity:
The quality here is high. They provide extensive experimental results to validate their idea, comparing TART with baselines across modalities and models. Every concept is backed up by a full sweep of experiments, they answer questions before I have a chance to ask them. For example extending this to vision and speech went above and beyond. The writing is clear well structured and the methodology is clearly explained. Their figures and tables are clear and accessible.

Significance:
I believe the significance is high. The authors address a limitation of LLMs with a novel solution that is broadly useful. It has potential to significantly improve transformer model's in context learning performance across the range of tasks.

Overall, great work.

**Weaknesses:**

There's a typo on line 339, it should be CIFAR-10 (classes plane and bird) and MNIST (classes 0 and 8)

**Questions:**

In a real world application of this technique it would be inconvenient for the user to have to indicate which tokens belong to which data points/labels, which I think the embedding averaging method requires. For example if I understand right
Sentence: The movie is good
Label: Positive
The user would have to manually indicate which tokens belong to the data (sentence) and label so it can be averaged.
I can imagine using the base model to separate them out for you. Is that how this would work?

**Limitations:**

Yes

---

> ### Author Rebuttal · Authors · 2023-08-10
>
> Thank you for the positive comments. We are glad that you found our work interesting on both the problem and the solution front, and found the paper well-written and clear.
>
> **Demarcating examples**. This is an interesting challenge. Currently, our code takes as input a demarcating limiter (which is “Sentence: and Label:” in the example you provided) and uses that to identify which positions to take the average embeddings over. If you are working with a fixed prompting template, implementing this is pretty straightforward. Alternatively, you could use Streaming embeddings (Figure 24) which side-steps this issue entirely because it embeds one example at a time.
>
> **CIFAR <-> MNIST.** Thanks for pointing out the minor typo. We will correct it in the updated draft.

---

### Official Review · Reviewer_hArx · 2023-07-07

**Soundness:** 3 good
**Presentation:** 2 fair
**Contribution:** 3 good
**Rating:** 6
**Confidence:** 3

**Summary:**

The paper studies why in-content learning achieves inferior performance compared with finetuning and adaptor, then proposes an LM-based inference module that learns to perform logistic regression based on the sample and previous (sample, label) sequence, where the sample and linear cutting-plane are sampled vectors. When testing, the base LLM encodes the in-content examples into single vectors, concatenates them with their corresponding labels to form the (sample, label) sequence and combines with the test sample to predict the test label. Experimental results show the proposed inference module can achieve comparable results without task-specific parameters.

**Strengths:**

The paper proposes a new way to classify a test sample based on training examples. The proposed inference module learns to predict the sample label given (x,y) examples, which share the same separation plane. By sampling enough various planes and input x, an LLM is possible to learn how to find the linear separation plane given (x,y) examples, thus correctly predicting the test label without further training. The proposed method may apply to meta-learning tasks as both settings are similar.

**Weaknesses:**

Although the proposed inference module is interesting, I can not see it has a significant advantage over linear probing.
Firstly, TART requires base LLM to encode examples and then fed the average pooled vector into the proposed inference module. This process requires a similar amount of computation and memory as in-context learning, and this process must repeat k times in the proposed LOO embedding if given k examples. On the contrary, linear probing can encode each sample independently and then train one linear layer for prediction.
Secondly, since the inference module is trained with linear logistic regression, the LLM representation must be linear separable for task-specific labels. Thus, it has the same upper bound as linear probing, while one linear layer should be much easier to train than a GPT-2 model.


Line 10 claims "performance gap exists due to their inability to perform simple probabilistic reasoning tasks" and Line 51, "...prompt engineering or active example selection, focus entirely on the LLM’s learned representations." The authors do not clearly explain what the reasoning ability is and why prompts improve learned representations. I assume prompt engineering is proposed due to the unavailability of hidden representation inside GPT-3 or GPT-4.

**Questions:**

What do the "parameters w" in Line 217 and "w_t" in Equation 3 means? Are these two refer to different things?

How does the TART obtain multi-way classification labels as it is trained with two-way classification? Can this method generalize to text generation with a much larger vocabulary?

Authors may compare TART with linear probing besides FT-layer given the same amount of examples.

**Limitations:**

The authors addressed the limitations and broader impact.

---

> ### Author Rebuttal · Authors · 2023-08-10
>
> Thanks for the detailed comments and feedback. We are glad that you find TART interesting and see applications of our work to settings beyond ICL (i.e., meta-learning).
>
> **Comparison with linear probing.** As we highlighted in the general response above, our main objective with this work is to understand why a gap exists between ICL and finetuning, and provide an algorithmic way to fix this gap. The caveat is to devise algorithms that can obtain the performance of finetuning while maintaining the benefits of ICL (namely, no need for training at inference time, providing a natural language interface to the user). As a first step towards this bigger goal, we begin by studying this problem for the class of binary classification tasks. While linear probing (for each task separately) would get close to finetuning performance, one has to forgo the advantages of ICL to use it. Furthermore, it is not possible to extend linear probing beyond the realm of classification tasks.
>
> On the other hand, TART allows users to benefit from the ICL framework and provides a “no training” way to match finetuning performance. Additionally, it opens up the possibility of improving the generative ability of LLMs using synthetic tasks. This is a promising line of work, which requires a new set of tools than those in the paper, and one which we are currently exploring.
>
> Additionally, we would like to clarify that the adapter head baseline, which is included in the original manuscript, is actually a slightly stronger baseline than the linear probing method. The adapter head comprises a single linear layer composed with a sigmoid nonlinearity. TART performs 3.4% better than these adapters.
>
> **Generalizing TART.** In the current form, you could obtain multiclass labels using a one-vs-all approach by looking at the logit probabilities for each class. However, to show that our method generalizes beyond binary classification, we trained an extension of TART for multiclass problems using multinomial logistic regression as the synthetic data generating task. We find that our TART inference module improves the base ICL performance by up to 59%. Please see Tables 1 and 2 in the attached PDF for details.
>
> **Explaining reasoning ability.** Throughout the paper, whenever we say the ability to perform reasoning we mean the ability to solve simple probabilistic reasoning problems, such as the logistic regression model. We make this connection mathematically precise in Equation (1) in  Section 2.3 (Line 149-150). The equation precisely defines what the representation and reasoning abilities are. The representation-reasoning decomposition in Eq. (1) allows us to formally state the hypotheses – H1, H2, and H3 —that we evaluate next.
>
> **Connecting prompt engineering with improved representations.** We agree that the connection between prompt engineering and improving representations was not elaborated on in the current version. We plan to revise and add this explanation: “Via prompt engineering users are searching the language space for a natural language template that best represents their task. This search enables the users to find the best representations for their specific task which makes the LLM more likely to solve it.” Thank you for pointing this out.
>
> **Minor typo.** The parameter $w$ in Line 217 should be $\theta$. We will fix this in the draft.

---

> > ### Comment · Reviewer_hArx · 2023-08-19
> >
> > I thank the authors for the response.
> >
> > My concerns about generalizing TART and Equation (3) have been addressed. But I agree with reviewer CKNq that the paper makes extraordinary claims about representation and reasoning. It requires much more evidence to support that the LLM can not perform simple reasoning.
> >
> > Without the claim, the proposed TART has its merits in improving ICL. It uses a GPT-2 to learn to approximate any linear transformation given examples (x_i,t, y_i,t). It is interesting and has similar concepts to meta-learning. However, it has several drawbacks: 1) You must sample enough w_t to span all possible distributions of the LLM output feature. 2) Why would one approximate the linear weights through extensive meta-learning instead of learning the linear weights, that is, linear probing straightly? With enough sampling and training costs, I think TART can outperform linear probing. But since Figure 5 and 6 shows the adapter surpass TART most of the time, I still have concerns about whether the merits are significant in practical use.
> >
> > Due to the above two points, I like to keep my rating unchanged but OK with both decisions.

---

> > > ### Author Response · Authors · 2023-08-19
> > >
> > > Thank you for your response and detailed feedback on our work. We are glad we were able to address your concerns on multi-class evaluations and Eq. (3). Please allow us to address the additional questions you asked.
> > >
> > > We agree with both you as well as Reviewer CKNq that our claims in Section 2.3 are restricted in scope to binary classification tasks and that it is not reflected in our writing. As we highlighted in our response to Reviewer CKNq that we will restrict the scope of those statements and make sure they are not taken out of context. In addition to this, we will make the analysis more general by extending it for multi-class classification problems as well.
> > >
> > > (From our response https://openreview.net/forum?id=ZXbgVm3PSt&noteId=Pc5b4H5FKn)
> > > > Reasoning vs. representation
> > >
> > > To make sure our claims are within the scope, we have updated:
> > > - Line 108: understand their relative performance for binary classification tasks.
> > > - Line 111: downstream binary classification tasks.
> > > - Line 144: This section investigates why this performance gap exists for binary classification tasks.
> > > - Line 151: insufficient reasoning abilities, i.e., the ability to learn linear decision boundary for a binary classification task
> > > - The hypothesis H1-H3 have been updated to focus on binary classification tasks.
> > > - Additionally, we will add an extra section in the appendix focussing on the same analysis with multi-class tasks.
> > >
> > > > On linear probing vs in-context learning
> > >
> > > Thank you for pointing out this question and we believe that the paper and our initial response failed to sufficiently outline the precise regime where TART is actually preferable to linear probing. We believe there are two regimes:
> > > - Single-task regime: In this regime, practitioners would like to improve the performance of a base model on a single task such as sentiment analysis. As your review highlights, linear probing is preferable to TART in this setting since it is computationally cheaper to optimize a linear model for a given task.
> > > - Multi-task regime: In this regime, practitioners would like to improve the performance of a base model over multiple tasks (e.g., all tasks on the RAFT benchmark) where in-context prompt is used to adapt the model for each task. This paradigm is reflected in numerous LLM benchmarks like OpenLLM leaderboard [1], HELM [2], and RAFT [3]. For these benchmarks, researchers have focussed on task-agnostic techniques like modifying pre-training data and fine-tuning the model on instruction/chat data. These are computationally expensive as they both require training the whole model. Our work argues that TART is intended for this regime. For instance, we show on the RAFT benchmark, TART improves GPT-NEO (125M)’s performance such that it outperforms BLOOM (176B), and is within 4% of GPT-3 (175B).
> > >
> > > The comparison to linear probing is to demonstrate that TART can match task-specific interventions while being task-agnostic. TARTs advantage over linear probing scales with the number of tasks that practitioners care about -- TARTs advantage over linear probing for one task is minimal, but the advantage for 100s of tasks (as captured by benchmarks and real-world user interactions with chat-models such as chatGPT) is substantial. This is because for each new task, linear probing requires practitioners perform task specific optimization (around 1 million parameters for sequence length 1024 and embedding dimension 1024).
> > >
> > > We are glad that you pointed out this question on "Why one cant one use linear probing" and we will update the manuscript with the above discussion distinguishing the two setups. We hope that this addresses your concerns and are happy to answer any more questions that you might have. We are really grateful for your response and insights.
> > >
> > > *References*
> > >
> > > [1] https://huggingface.co/spaces/HuggingFaceH4/open_llm_leaderboard
> > >
> > > [2] Liang, Percy, Rishi Bommasani, Tony Lee, Dimitris Tsipras, Dilara Soylu, Michihiro Yasunaga, Yian Zhang et al. "Holistic evaluation of language models." arXiv preprint arXiv:2211.09110 (2022).
> > >
> > > [3] Alex, Neel, Eli Lifland, Lewis Tunstall, Abhishek Thakur, Pegah Maham, C. Jess Riedel, Emmie Hine et al. "RAFT: A real-world few-shot text classification benchmark." arXiv preprint arXiv:2109.14076 (2021).

---

> > > > ### Comment · Reviewer_hArx · 2023-08-21
> > > >
> > > > Thanks for the detailed response on comparing with linear probing. It has addressed my concerns. I will raise my rating to WA.

---

> > > > > ### Author Response · Authors · 2023-08-21
> > > > > **Thanks**
> > > > >
> > > > > Thank you for questions and feedback. They have helped us improve our draft significantly. We really appreciate your time and effort!

---

### Official Review · Reviewer_VGrP · 2023-07-10

**Soundness:** 2 fair
**Presentation:** 2 fair
**Contribution:** 2 fair
**Rating:** 4
**Confidence:** 3

**Summary:**

The paper presents an recipe for adapting an LLM to perform classification tasks in a task agnostic manner. They first try to tease apart if existing "in-context" methods which construct prompts to describe the task and then ask for the inference result are not achieving great performance because of information extraction or reasoning ability. By constructing a linear probe experiment they get indications that it may be the reasoning ability which hampers performance. The authors then propose to improve the task performance by training an inference module to solve a generic task, here few shot logistic regression, and then show that they can apply this to the outputs of the LLM to obtain good performance on desired tasks. The down stream tasks considered range from sentiment classification, news article categorization to spam detection. They also verify that the performance gains on text tasks carries over to image tasks and speech.

**Strengths:**

* There is still a lot of scope and practical use in extracting information from a text model in a reliable way.
* The method is novel in so far I can tell.
* Contains reasonable baselines.

**Weaknesses:**

* Although I find it intriguing the paper leaves me more questions than it answers. I think it would be very beneficial to make the presentation much more concrete and map some of the down stream tasks clear and exemplified in the main text. They remain very abstract and lots of intuition that the readers should build up is lost that way.
* Although I think the line of scientific inquiry is quite intriguing I don't find the paper mature enough to be published in the current form. I find the writing can be improved a lot and maybe the context can be expanded a bit.

**Questions:**

* Th1 seems to say that the error is controlled by how similar P_NL and P_syn are but they are really different. In some sense they should always be different because it's a different language with a very different structure. In NL we have a lot more expressivity and less regularity ? Did you do any experiment that broadens the task space to see if this improves ?
* I think 3.3 should be more appropriately named how to leverage LLM embeddings. The embeddings are always the last layer it's more how they are used that changes ?
* Are all the tasks considered binary classifications ? I guess line 339 should be corrected MNIST and CIFAR are switched. We can't easily evaluate this work when this is happening.

**Limitations:**

None.

---

> ### Author Rebuttal · Authors · 2023-08-10
>
> Thank you for your thoughtful comments on our work and we are glad you found our proposed method novel. First, we address your major concern regarding the presentation of our work, and then respond to your questions.
>
> **Quality of presentation.** While other reviewers (tMpn, CKNq) found the quality of our presentation to be high, we welcome your feedback on how it can be further improved. We will make the following changes to the final camera-ready draft to improve its readability:
> - Update the introduction section to better reflect our objectives and motivate our proposed method. We will use the outline from the general response to communicate our message better.
> - Add an additional section in the main body (derived from content from Appendix A) on using synthetic finetuning to improve a pre-trained model's ICL ability.
> - We will add some more details of our downstream evaluation tasks in language, vision and speech (expanding on lines 67-73). Currently, these are detailed in Section 4 with details deferred to Appendix D due to lack of space.
> - For Section 3.3, our choice of name “Which embeddings to take” was to reflect the fact that one can use the LLM in different ways to embed the input sequences (i.e.,  Vanilla embeddings or leave-one-out (LOO) embeddings). We agree that it might be more appropriate to focus on “How to embed” as opposed to “Which embeddings to take”. We will make this change.
> - We will fix the typos in the revised draft (MNIST <-> CIFAR).
>
> **Closeness of distributions P_NL and P_SYN in Theorem 1.** We would like to clarify that P_{NL} is the distribution induced on the embeddings of the natural language data rather than the data itself (see Lines 619-622). Furthermore, when we combine the TART module with the base LLM, we perform a PCA renormalization step (Lines 221-224) which projects the embeddings into the same space as the synthetics. These two facts combined ensure that P_syn and P_NL are indeed very similar to each other in our experiments. It is indeed surprising that projection+renormalized NL embeddings are close to the synthetic Gaussian distribution. We will add this discussion to Section 3.4 to make it clear.
>
> **Extensions of TART beyond binary classification.** In this work, we focus on the set of binary classification tasks across language, vision and speech domains. That said, based on the feedback from the reviewers, we provide additional experiments (see Table 1 and Table 2 in attached PDF) demonstrating that our method can indeed be extended to multiclass tasks, improving base ICL performance by up to 59%.

---

### Author Rebuttal · Authors · 2023-08-10

We thank all the reviewers for their thoughtful comments and feedback. We are glad that they found TART to be **novel** and **practical** (VGrP, CKNq, tMpn), **well-motivated** (CKNq, tMpn), and supported by **extensive experiments** (VGrP, tMpn, CKNq).

Recall that our paper shows that in-context learning (ICL) is much more competitive with finetuning than previously thought. Further, we show that this is achievable by training an independent Transformer module entirely on synthetic logistic regression data.

Based on the feedback, we clarify two points on our presentation quality and scope of TART. We will make the following small, yet important, changes to our paper:
- **Presentation.** Our work presents two approaches to improving the ICL performance: (1) finetuning the entire model with synthetic binary classification data and (2) training an independent module (TART) on synthetic data. Following VGrP and hArx’s comments on presentation quality, we realized we had underemphasized point (1), which is a natural first step towards TART. Additionally, this point actually demonstrates that pretrained models do not learn how to classify in-context from standard pretraining objectives. The experiments for (1) were mentioned briefly in the main body (deferred to App. A) and we will update the draft with an additional section incorporating this.
- **Scope of TART.** As a result of the questions posed by tMpn and CKNq, we realized that our TART module could be generalized in two useful ways: (1) to multiclass classification problems, and (2) in the low-data regime (4-10 examples). We conducted additional experiments (see attached PDF) and show that for multiclass problems, TART outperforms base ICL performance by up to 59%, far exceeding the gains we saw on binary problems. Further, in the low-data regime, it can be up to 31% better than in-context LLM accuracy and outperforms finetuning on 2/3 datasets.

---
**Detailed response**

We synthesize all major reviewer concerns and considerations, and show how we incorporate these in our presentation either through minor rewrites or with additional experiments.

Our paper is about understanding why the ICL ability of pretrained LLMs has sub-par performance when compared with SGD-based finetuning. ICL has fundamentally changed how people use AI systems by providing a new natural language interface for specifying tasks. Yet, it still lags finetuning methods in performance quality (up to 30% on binary classification). Our objective is to understand whether this performance gap is an inherent limitation of ICL or not.
- hArx suggests linear probing as an alternative. While we agree that it might be competitive with finetuning in several cases, it doesn’t address the main question that we consider: why is ICL under-performing finetuning? Additionally, linear probing takes away ICL’s intuitive interface from a user’s interaction with an LLM.

We begin by asking why this quality gap exists: do the data representations learned by the models lack task-specific information? Or is there some deficiency in the model’s ability to use this information, i.e., reasoning ability? For classification tasks, our results suggest that pretrained LLMs indeed have the necessary information in the representations, but fail because of their inability to do simple forms of reasoning, e.g. linear classification.
- We agree with CKNq that understanding this question is a challenging study to undertake, with little agreement in the literature. Our representation-reasoning decomposition is a first step towards formalizing this question (Eq. 1 provides a mathematical description). While Sec. 2.3 currently focuses on binary tasks, we will perform a similar analysis for multiclass tasks.

As a first step towards improving ICL, we finetuned an LLM with synthetic binary classification problems – observe that we are updating the weights of the model here. While this procedure improves model performance on classification tasks by up to 17% (see App. A), such finetuned models might lose their generative capabilities. This brings about the question: is it possible to improve the ICL abilities of these models without interfering with its other pre-existing capabilities?
- To partially address VGrP and hArx’s concern on presentation, we will add a section on this synthetic finetuning between Sec. 3 & 4.

We undertook this challenging task of teaching a model a new generic skill without affecting existing capabilities. Rather than directly finetuning, we trained an independent Transformer module (TART) entirely on synthetic data to learn this classification skill. By applying this module to the output embeddings of an arbitrary pretrained LLM, we improved the model’s ICL ability to be competitive with finetuning (within 3% from a gap of over 20%). Additionally, this surprising finding demonstrates that pretrained models do not learn how to classify (or reason over embeddings) in the pretraining phase.

While our work focuses on binary classification, we recognized (following tMpn and CKNq’s suggestion) that our procedure is not restricted to binary tasks. We trained a multiclass version of the TART module and our preliminary results suggest that it can outperform base ICL by up to 59%, more than the improvements on binary tasks. On CKNq’s suggestion, we also explored TARTs performance in the low-data regime (4-10 examples). Excitingly, we found that it can be up to 31% better than base in-context LLM and that it even outperforms finetuning on 2/3 datasets in that data regime (see attached PDF).
- We will add detailed experiments for multiclass tasks and extend our current evaluations to the low-data regime as suggested by tMpn and CKNq.

Further, we go beyond text and show that the same TART module can be combined with vision and speech models to enable ICL for these modalities. This suggests that this capability is somehow uniformly lacking across a range of foundation models.

---

---

### Decision · Program_Chairs · 2023-09-21

**Decision:**

Accept (poster)

**Comment:**

This paper studied the gap of performance between in context learning and the task-specific tuning.
The reviewers generally like the main insight of decoupling the representation and reasoning, but had concerns regarding the presentation, generality and the effectiveness of the proposed method. During the rebuttal, the authors have done a good job in addressing the most of the concerns, so I would recommend accepting the paper, with the condition that the authors would update the paper accordingly to reflect the changes in their camera ready version.